# Unlocking Foundation Models for Time Series with Channel Descriptions

## Abstract

Traditional time series models are often task-specific and rely heavily on feature engineering. While Transformer-based architectures have advanced sequence modeling in other domains, their use for time series representation learning remains limited. We introduce CHARM, a model that improves representation quality for multivariate time series by incorporating channel-level textual descriptions into its architecture. This design enables the model to exploit contextual information associated with individual sensors while remaining invariant to channel order. CHARM is trained using a Joint Embedding Predictive Architecture (JEPA) with a novel loss function that encourages informative and temporally robust embeddings. We find that integrating channel descriptions consistently enhances representation quality, with supplementary ablations providing insight into the contributions of different design choices. The learned embeddings yield strong performance across diverse downstream tasks, underscoring the value of description-aware time series modeling.

## 1 Introduction

Time series models play a pivotal role in critical real-world applications such as forecasting, classification, and anomaly detection across domains including manufacturing, energy, healthcare, and finance (Hannun et al., 2019; Susto et al., 2014; Ding et al., 2015). By converting temporal signals into actionable insights, these models enable large-scale, data-driven decision-making. However, most existing approaches remain narrowly scoped and task-specific, requiring significant manual effort for development and maintenance. Even in ostensibly homogeneous settings—such as industrial pump fleets with varied sensor configurations—models are often trained independently (Morgenthal et al., 2024), despite underlying shared physical dynamics. This fragmentation is rooted in structural limitations of conventional time series architectures, which typically assume fixed-length, uniformly structured inputs and lack mechanisms for fusing information across heterogeneous sensors. Consequently, current paradigms struggle to generalize across tasks, domains, and configurations, posing challenges to scalability and adaptability.

**Foundation models in other modalities** In contrast, fields such as natural language processing, computer vision, and audio have undergone transformative progress with the emergence of foundation models—large-scale, pre-trained architectures that learn general-purpose representations across diverse downstream tasks (Devlin et al., 2019; Nussbaum et al., 2025; Assran et al., 2023; Kirillov et al., 2023; Baevski et al., 2020; Brown et al., 2020; Radford et al., 2021). These models, often trained via Self-Supervised Learning (SSL) on massive unlabeled corpora, have demonstrated capabilities such as Retrieval-Augmented Generation (RAG) (Lewis et al., 2020) and robust task transfer via lightweight fine-tuning (Devlin et al., 2019; Oquab et al., 2023; Kirillov et al., 2023). Their success hinges on learning semantically meaningful representations that are modular, robust, and highly transferable.

**Foundation models for time series forecasting** Inspired by these advances, the time-series community has begun developing foundation models, with a strong emphasis on supervised forecasting objectives (Das et al., 2024; Woo et al., 2024b; Ansari et al., 2024; Liu et al., 2024). These models achieve impressive performance on predictive benchmarks and introduce architectural innovations tailored to multi-domain forecasting. However, because their training remains tightly coupled

to a forecasting loss, the learned representations are often specialized and brittle—limiting their applicability to downstream tasks such as classification, segmentation, or anomaly detection.

**Foundation embedding models for time series**   Most self-supervised foundation models for time series rely on objectives such as masked reconstruction or next-step forecasting, which require the encoder to impute raw signal values. These signals are often noisy, low-resolution, and entangled with domain-specific artifacts (Trirat et al., 2024), resulting in representations that overfit to sensor-level noise rather than capturing higher-level process dynamics. While such objectives are straightforward to implement, they tend to entangle semantic structure with noise, limiting robustness and generalization across tasks or domains. Recent approaches such as MOMENT (univariate) (Goswami et al., 2024) and UniTS (multivariate) (Gao et al., 2024a) extend this paradigm with multi-task or reconstruction-based pretraining and report strong downstream performance, but remain fundamentally grounded in raw signal-level prediction for pretraining.

**JEPA-style latent prediction: a robust alternative**   In contrast, Joint Embedding Predictive Architectures (JEPA) (LeCun, 2022) adopt a fundamentally different approach: predicting latent representations of masked target segments from contextual embeddings rather than raw values. By operating entirely in embedding space, JEPA filters out sensor noise and encourages the encoder to model higher-level temporal structure. In vision, this paradigm has proven highly effective—Assran et al. (2023) demonstrate that latent prediction yields representations that rival or surpass those learned via supervised learning, while remaining more robust to noise and label scarcity. Compared to contrastive learning, JEPA-style models also avoid the complexity of negative sampling and the sensitivity to embedding space dimensionality, making them a more stable and scalable choice for semantic representation learning.

**Lack of channel-awareness in time series models**   Most time series models treat all input channels uniformly as uncategorized streams of sensor data, without incorporating information about the identity, modality, or semantics of the sensors generating the data. This lack of sensor-awareness discards valuable contextual information, limiting the model's ability to reason about sensor-specific behavior or operate reliably across deployments with varying instrumentation.

## 1.1 CONTRIBUTIONS

This work aims to (1) develop a robust, semantically grounded SSL objective for time series data, and (2) design an architecture capable of directly incorporating textual channel information. This is inspired by how subject matter experts interpret time series data, by jointly considering the raw signals and their accompanying channel descriptions. To this end, we introduce a **CH**annel-**A**ware **R**epresentation **M**odel (CHARM), trained to produce domain-aware, and performant representations across tasks and datasets. Building such a model entails several key challenges, including channel heterogeneity, variation in temporal dynamics across domains, and risks of negative transfer and representational collapse. To realize these aims, we introduce the following core contributions:

**Description-aware temporal featurization**   We modify temporal convolutional networks to incorporate channel descriptions directly into the convolutional layers. Unlike patch-based approaches, our stacked, description-aware convolutions allow the model to seamlessly adapt across domains without manual tuning of patch size. Details are provided in Section 2.1.1.

**Inter-channel reasoning via attention and gating**   We augment the standard attention mechanism with novel, learnable inter-channel attention layers and gating modules conditioned on channel descriptions. These components enable the model to flexibly capture inter-channel dependencies, selectively integrate signals in a structured manner, while maintaining invariance to channel ordering. See Section 2.1.2 for details.

**Self-supervised training with JEPA for time series**   We adapt the JEPA to the time series domain, enabling semantic representation learning without reconstruction. To do so, we introduce a set of tailored data augmentations and temporal perturbations that improve robustness to common time series artifacts. This avoids the drawbacks of contrastive learning, such as sensitivity to sampling and

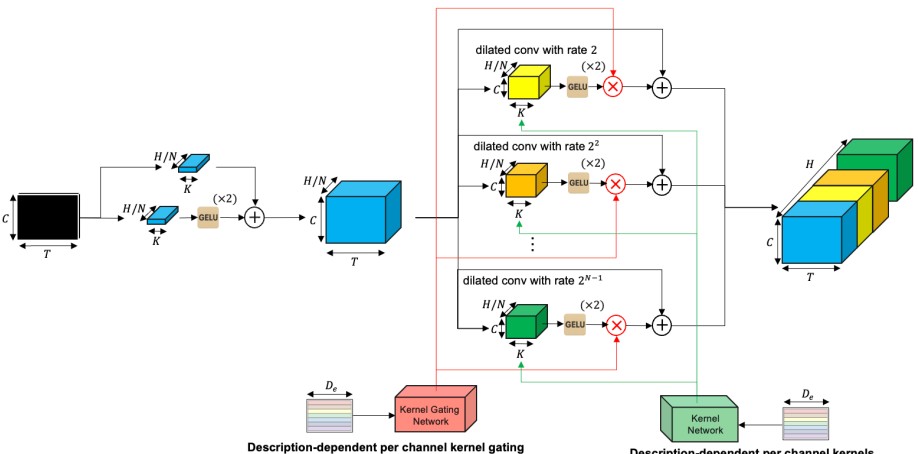

**Figure 1:** Overview of the model architecture, featuring a context-aware temporal convolutional network and a series of contextual attention layers, each guided by textual descriptions of the input time series channels.

**Figure 2:** Schematic of the context-aware temporal convolutional network, performing initial featurization of multivariate time series inputs guided by granular textual descriptions of each channel.

dimensionality constraints (LeCun, 2022; Assran et al., 2023; Chen et al., 2020; 2022; Chuang et al., 2020). Details are provided in Section 2.2.

We evaluate our model across a range of downstream tasks, including classification, forecasting, and anomaly detection. Our approach consistently achieves strong performance across diverse datasets, underscoring the effectiveness of both the model architecture and the training strategy.

## 2 METHODOLOGY

In this section, we first introduce a novel multi-modal transformer-based architecture for learning embeddings from time series data, guided by underlying channel descriptions (Section 2.1). We then describe how this architecture is trained using self-supervised learning with JEPA (Section 2.2). The notation used throughout this section is provided in Appendix B.

### 2.1 MULTI-MODAL TIME SERIES EMBEDDING MODEL

Here, we present three key architectural contributions that enable learning high-quality time series embeddings by incorporating textual channel descriptions. Our model employs convolutional layers in conjunction with a series of custom attention layers, enhanced by a novel attention mechanism. An overview of the full architecture is provided in Figure 1.

We begin by describing the contextual temporal convolutional network in Section 2.1.1, which generates convolution-based embeddings. These embeddings are then passed to a series of contextual attention layers, where our novel attention mechanism is applied. We describe the details of this layer in Section 2.1.2, where we introduce two core extensions to the self-attention mechanism in sections.

### 2.1.1 CONTEXTUAL TEMPORAL CONVOLUTIONAL NETWORK

We introduce a contextual Temporal Convolutional Network (TCN) that projects input time series $\mathbf{T} \in \mathbb{R}^{T \times C}$ into contextual embeddings $\mathbf{T}_c \in \mathbb{R}^{T \times C \times H}$, where $\mathbf{T}_c[i, j, :]$ denotes the $H$-dimensional embedding at time step $i$ for channel $j$. The base architecture follows standard dilated TCNs (Bai et al., 2018; Lin et al., 2021), which stack 1D convolutions with exponentially increasing dilation factors ($2^l$). However, standard TCNs are architecturally static and their learned kernels are input independent and constant. This lack of flexibility hinders their ability to adapt across diverse domains, leading to representation collapse or negative transfer when trained on heterogeneous datasets. To address this, we make the TCN *context-aware* by incorporating channel descriptions into the convolutional process. Given a time series tuple $\mathbf{t} = (\mathbf{T}, \mathbf{D}, \mathbf{pos})$ (see Figure 7), we extract text embeddings for the descriptions using a frozen text embedding model, as $\mathbf{E}_d \in \mathbb{R}^{C \times D_e}$. We introduce two mechanisms to inject this context, namely:

**Contextual kernel gating** Description embeddings are used to conduct soft gating through the layers of the TCN. The gates are produced by the *kernel gating network* in Figure 2, which is given as $\mathbf{G}_c = \mathbf{sigmoid}(\mathbf{E}_d \mathbf{W}_g), \mathbf{W}_g \in \mathbb{R}^{D_e \times N}$, with $N$ denoting the number of stacked convolutional layers in the TCN. Each element $\mathbf{G}_c[i, j]$, which corresponds to the soft gate associated with channel $i$ and layer $j$ of the TCN which is then incorporated multiplicatively in the network as depicted in Figure 2. This enables the model to control the effective field of view of TCN informed by the channel descriptions.

**Contextual kernels** Rather than learning fixed convolutional filters, we generate them from the descriptions embeddings as $\mathbf{G}_k = \mathbf{E}_d \mathbf{W}_k, \mathbf{W}_k \in \mathbb{R}^{D_e \times (\frac{H \times K}{N})}$, where $K$ is the kernel size and $N$ the number of TCN layers. This mechanism directly ties channel semantics to filter generation and is represented by the *kernel network* in Figure 2.

### 2.1.2 CONTEXTUAL ATTENTION LAYER

The embeddings generated by the contextual TCN layer are subsequently processed through a sequence of contextual attention layers. The primary goal of these layers is to effectively fuse channel and temporal dimensions into richer, more expressive representations, directly incorporating the granular textual descriptions of each channel. To achieve this, we propose several novel extensions to the classical self-attention mechanism (Vaswani et al., 2017). These and their inter-play are depicted in Figure 1, under the contextual attention layer. Below we discuss the key details of these key components in detail.

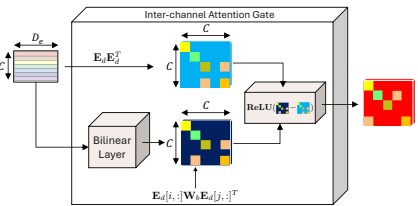
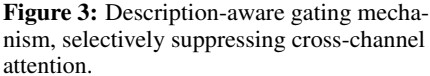
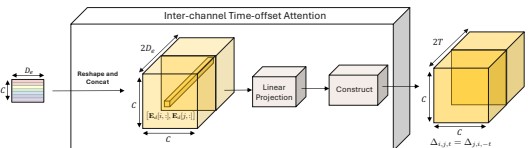

**Figure 3:** Description-aware gating mechanism, selectively suppressing cross-channel attention.

**Figure 4:** Symmetric construction of inter-channel temporal-offset attention, encoding mutual dependencies between channels at varying time lags.

**Description-aware inter-channel attention gating** This module introduces a gating mechanism conditioned explicitly on channel descriptions, enabling our architecture to selectively co-attend to the most relevant channels. Given the channel description embeddings $\mathbf{E}_d$, this layer computes the pairwise similarities, $\mathbf{S}$, and the similarity threshold matrix, $\mathbf{Z}$, as $\mathbf{S} = \mathbf{E}_d \mathbf{E}_d^\top$, $\mathbf{Z}[i, j] = \mathrm{sigmoid}\left(\mathbf{E}_d[i, :] \mathbf{W}_b \mathbf{E}_d[j, :]^\top\right)$. The similarity threshold matrix governs our inter-channel gating mechanism. Specifically, this layer outputs the gating matrix given as $\mathbf{G}_d = \mathrm{ReLU}(\mathbf{Z} - \mathbf{S})$. This process, illustrated in Figure 3, allows the model to selectively suppress cross-attention between channel pairs $(i, j)$ by driving their corresponding similarity threshold $\mathbf{Z}[i, j]$ toward one.

**Description-aware inter-channel time-offset attention** This module improves the model's ability to capture inter-channel relationships by explicitly quantifying dependencies between channels at different temporal offsets. Specifically, we introduce a learnable tensor $\mathbf{\Delta} \in \mathbb{R}^{C \times C \times 2T_{\max}}$, where each entry $\mathbf{\Delta}_{i,j,t}$ encodes the learned dependency strength between channel $i$ and channel $j$ at a temporal offset of $t$ steps. We assume inherent symmetry within $\mathbf{\Delta}$, reflecting the intuition that the relationship from channel $i$ to channel $j$ at step $t$ should match the inverse relationship from channel $j$ to channel $i$ at step $-t$, formally as $\mathbf{\Delta}_{i,j,t} = \mathbf{\Delta}_{j,i,-t}$. To explicitly enforce this symmetry, we follow a structured construction procedure. Given the channel description embeddings $\mathbf{E}_d$, we first create a pairwise embedding tensor $\bar{\mathbf{E}}_d \in \mathbb{R}^{C \times C \times 2D_e}$ defined by concatenation as $\bar{\mathbf{E}}_d[i,j,:] = [\mathbf{E}_d[i,:], \mathbf{E}_d[j,:]] \in \mathbb{R}^{2D_e}$.

Next, we apply a linear projection to these pairwise embeddings, parameterized by the matrix $\mathbf{W}_d \in \mathbb{R}^{2D_e \times T_{\max}}$, yielding the intermediate tensor $\mathbf{\Delta}_+ = \bar{\mathbf{E}}_d \mathbf{W}_d$. We then construct the full symmetric tensor $\mathbf{\Delta}$ as below

$$
\mathbf{\Delta}[i,j,t] = \begin{cases} \mathbf{\Delta}_+[i,j,t] & \text{if } t \geq 0, \\ \mathbf{\Delta}_+[j,i,-t] & \text{if } t < 0. \end{cases}
$$

This symmetric construction, depicted in Figure 4, ensures parameter efficiency and explicitly encodes symmetry constraints. We compute the final $\bar{\mathbf{\Delta}} \in \mathbb{R}^{CT \times CT}$ matrix using a "slice-and-tile" operation, where $\bar{\mathbf{\Delta}}$ is a block matrix with $T$ blocks on each axis, and in block notation $\bar{\mathbf{\Delta}}[T_i, T_j] = \mathbf{\Delta}[:, :, T_j - T_i]$. See Section C.5.1 for PyTorch style pseudocode for the naive and fast versions of this operation.

**Custom attention mechanism** We unify the gating and attention mechanisms described above into a single self-attention framework. Given embedding matrix the contextual TCN layer, $\mathbf{T}_c$, we reshape it into $\mathbf{X} \in \mathbb{R}^{CT \times H}$, where each channel-time pair is represented by an $H$-dimensional embedding. To facilitate intuitive indexing, we employ a triple-index notation $\mathbf{X}_{[(c_i, t_j), k]}$ rather than a flattened indexing scheme $\mathbf{X}[m, k]$, with $c_i = m \mod C$ and $t_j = \lfloor \frac{m}{C} \rfloor$. First we apply rotary position embeddings to the queries and keys given the **pos** indices as:

$$
\hat{\mathbf{Q}} = \text{RoPE}(\mathbf{W}_Q \mathbf{X}_{[(i,p),:]}, \mathbf{pos}), \quad \hat{\mathbf{K}} = \text{RoPE}(\mathbf{W}_K \mathbf{X}_{[(j,q),:]}, \mathbf{pos})
$$

The custom attention matrix $\mathbf{A} \in \mathbb{R}^{CT \times CT}$ is then constructed as

$$
\mathbf{A}_{[(i,p),(j,q)]} = \text{Softmax} \left( \underbrace{\frac{\hat{\mathbf{Q}}_{[i,p,:]} \hat{\mathbf{K}}_{[j,q,:]}^T}{\sqrt{D_e}}}_{\text{Vanilla Self-Attention}} + \underbrace{\mathbf{\Delta}[i,j,q-p]}_{\text{Channel Lags}} - \underbrace{\lambda_G \mathbf{G}_d[i,j]}_{\text{Channel Gates}} \right)
$$

Here, $\mathbf{A}_{[(i,p),(j,q)]}$ represents the attention from channel $i$ at time $p$ to channel $j$ at time $q$. The scalar $\lambda_G$ is typically a large positive number, enabling the gating matrix to serve as an attention mask, selectively blocking certain cross-channel interactions based on the learnt thresholds. The attention matrix can be efficiently computed using vectorized operations by appropriately tiling the inter-channel gating and time-offset matrices. Following the standard transformer approach, we multiply the attention matrix by the value matrix $\mathbf{V} = \mathbf{W}_V \mathbf{X}$ to produce our contextualized embeddings.

### 2.1.3 Putting It All Together

This completes the integration of the various components within our multimodal time-series embedding architecture. For a given input tuple $\mathbf{t} = (\mathbf{T}, \mathbf{D}, \mathbf{pos})$, we first generate the initial embeddings $\mathbf{X} \in \mathbb{R}^{T \times C \times H}$ from our **contextual TCN** layer. These embeddings pass through a stack of $N$ **contextual attention** layers, each layer outputting $\mathbf{X}^{(l)} \in \mathbb{R}^{(T \times C) \times H}$, reshaped to $\mathbf{Y}^{(l)} \in \mathbb{R}^{T \times C \times H}$ for subsequent layers. Similar to (Grill et al., 2020a), we apply $\ell_2$ normalization to the final embeddings: $\mathbf{Y}[i,j,t] = \frac{\mathbf{Y}[i,j,t]}{\sqrt{\sum_h \mathbf{Y}[i,j,h]^2}}$, with normalization computed along the embedding dimension only. The complete architecture is denoted as $\mathbf{E}_\theta$, such that $\mathbf{Y} = \mathbf{E}_\theta(\mathbf{T}, \mathbf{D}, \mathbf{pos})$. While this outlines the primary structure of our contextual embedding model, we have also introduced several nuanced modifications aimed at enhancing training stability and convergence speed. These detailed adjustments are presented in Section C.

## 2.2 SELF-SUPERVISED REPRESENTATION LEARNING

We adopt the JEPA framework (Assran et al., 2023; LeCun, 2022) to enable self-supervised learning on time series data enriched with fine-grained textual context. JEPA comprises three core components, namely predictor, context, and target encoders. In the following section, we detail the key components of our training pipeline based on JEPA, namely, (i) the dataset generation process in Section 2.2.1, (ii) the integration of JEPA with our embedding model in Section 2.2.2, and (iii) a novel loss formulation tailored to JEPA training in Section 2.2.3.

### 2.2.1 DATASET GENERATION

Figure 5 provides a high-level view of JEPA and the interplay among JEPA's three encoders, and how they consume data points generated by the dataset generation process. For time series data, we generate augmented views of the same data point through data augmentation and perturbation techniques. We refer to the data augmentation as JEPA training tasks.

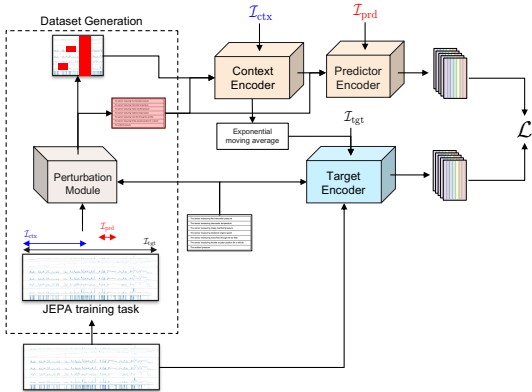

**Figure 5:** JEPA architecture with three encoders processing augmented views.

Formally, given an input instance $\mathbf{t} = (\mathbf{T}, \mathbf{D}, \mathbf{pos})$, we define an augmented view of this instance through three randomly generated contiguous sets of indices, with $\mathcal{I}_{ctx}, \mathcal{I}_{tgt}, \mathcal{I}_{prd} \subseteq \mathbf{pos}$, denoting time indices that are fed into the context, target and predictor encoders. We employ two self-supervised tasks, namely *causal prediction*, where $\mathcal{I}_{ctx} \subset \mathcal{I}_{tgt}$, and *smoothing*, where $\mathcal{I}_{ctx} \cap \mathcal{I}_{tgt} \neq \emptyset$ and $\mathcal{I}_{prd} \subset \mathcal{I}_{ctx} \cap \mathcal{I}_{tgt}$. See Figure 10 for an overview of the causal prediction (left) and smoothing (right) tasks. The input to the context encoder is further perturbed (see Section I.1) to encourage the model to learn robust representations under mild corruption.

### 2.2.2 JEPA SETUP

In JEPA the context and target encoders are architecturally identical. However, only the context encoder is directly optimized during training while the target encoder is updated using an exponential moving average of the context encoder's parameters, see Figure 5. In contrast, the predictor is a narrow/shallower version of the context encoder which is trained jointly with the context encoder through standard backpropagation.

To leverage JEPA, we integrate our embedding model within the underlying encoders. Let us denote the context, target and predictor encoders as, $\mathbf{E}_\theta^c, \mathbf{E}_\theta^t$ and $\mathbf{E}_\theta^p$, respectively. The context and target encoders are fully defined by our embedding model as

$$\mathbf{X}^c = \mathbf{E}_\theta^c(\bar{\mathbf{T}}, \bar{\mathbf{D}}, \mathcal{I}_{ctx}) \coloneqq \mathbf{E}_\theta(\bar{\mathbf{T}}, \bar{\mathbf{D}}, \mathcal{I}_{ctx}), \quad \mathbf{X}^t = \mathbf{E}_\theta^t(\mathbf{T}[\mathcal{I}_{tgt}, :], \mathbf{D}, \mathcal{I}_{tgt}) \coloneqq \mathbf{E}_\theta([\mathcal{I}_{tgt}, :], \mathbf{D}, \mathcal{I}_{tgt})$$

where $\mathbf{X}^c \in \mathbb{R}^{|\mathcal{I}_{ctx}| \times C \times H}$ and $\mathbf{X}^t \in \mathbb{R}^{|\mathcal{I}_{tgt}| \times C \times H}$ are the outputs from the context and target encoders, respectively, and $\bar{\mathbf{D}}$ and $\bar{\mathbf{T}} \in \mathbb{R}^{|\mathcal{I}_{ctx}| \times C}$ denote the perturbed descriptions and the perturbed time series data $\mathbf{T}[\mathcal{I}_{ctx}, :]$. The predictor encoder accepts the output of the context encoder as input. Unlike the context and target encoders, the predictor encoder solely leverages the contextual attention layer. Let $\bar{\mathbf{A}}_\theta$ denote the narrower and shallower version of the contextual attention layer. Also let $\bar{\mathbf{X}}_c = [\mathbf{X}^c, \underbrace{\mathbf{m}_\theta, \cdots, \mathbf{m}_\theta}_{\text{repeated } |\mathcal{I}_{prd}| \text{ times}}]$ where $\mathbf{m}_\theta$ represents learnable placeholders that guide the predictor encoder to generate embeddings for masked positions, see (Assran et al., 2023) for more information. We further define the concatenated set $\bar{\mathcal{I}}_{prd} = \mathcal{I}_{ctx} + \mathcal{I}_{prd}$. The predictor encoder is then defined as $\mathbf{X}^p = \mathbf{E}_\theta^p(\bar{\mathbf{X}}^c, \bar{\mathbf{D}}, \bar{\mathcal{I}}_{prd}) \coloneqq \bar{\mathbf{A}}_\theta(\bar{\mathbf{X}}^c \mathbf{W}_{pd}, \bar{\mathbf{D}}, \bar{\mathcal{I}}_{prd}) \mathbf{W}_{pu}$, where $\mathbf{X}^p \in \mathbb{R}^{|\bar{\mathcal{I}}_{prd}| \times C \times H}$ is the output of the predictor encoder, and $\mathbf{W}_{pu} \in \mathbb{R}^{H_d \times H}$, $\mathbf{W}_{pd} \in \mathbb{R}^{H \times H_d}$ denote linear layers used for up and down projecting.

### 2.2.3 TRAINING LOSS

The training loss for training our embedding model comprises two major components, self-supervised objectives and regularization terms associated with key modules of the contextual attention layer.

**Self-supervised loss**  Our embedding model produces embeddings at the level of each time point and each channel. We employ a self-supervised objective based on the $\ell_1$ norm to measure discrepancies between embeddings from two augmented views of the same time series instance. To promote consistency not only at the most granular level but also across coarser aggregations, we extend the objective to include progressively aggregated embeddings. Let $\bar{\mathbf{X}}^t = \mathbf{X}^t[\mathcal{I}_{\mathrm{prd}}, :] \in \mathbb{R}^{|\mathcal{I}_{\mathrm{prd}}| \times C \times H}$ and $\bar{\mathbf{X}}^p = \mathbf{X}^p[-|\mathcal{I}_{\mathrm{prd}}| :, :, :] \in \mathbb{R}^{|\mathcal{I}_{\mathrm{prd}}| \times C \times H}$. The self-supervised loss is then defined as

$$\mathcal{L}_{\mathrm{ssl}} = \sum_{i,j,t} \left| \bar{\mathbf{X}}^p_{i,j,t} - \bar{\mathbf{X}}^t_{i,j,t} \right| + \sum_{i,t} \left| \mu_j \left( \bar{\mathbf{X}}^p_{i,:,t} \right) - \mu_j \left( \bar{\mathbf{X}}^t_{i,:,t} \right) \right| + \sum_{t} \left| \mu_{i,j} \left( \bar{\mathbf{X}}^p_{:,:,t} \right) - \mu_{i,j} \left( \bar{\mathbf{X}}^t_{:,:,t} \right) \right|$$

with $\mu_j(\mathbf{X}_{i,:,t}) = \frac{1}{C} \sum_j \mathbf{X}_{i,j,t}$, $\quad \mu_{i,j}(\mathbf{X}_{:,:,t}) = \frac{1}{CT} \sum_i \sum_j \mathbf{X}_{i,j,t}$. This multi-resolution loss encourages the model to align representations both at the fine-grained level (per time point and channel) and at higher levels of abstraction (per time point and globally), thereby enhancing regularity and usability of the embeddings at different levels of granularity.

**Regularization loss**  We include two regularization terms related to key modules of the contextual attention layer, namely the inter-channel gating and inter-channel time-offset attention modules. Given the inherent sparsity in meaningful channel relationships, we promote sparsity in the learned channel relationships, by regularizing the similarity threshold matrix $\mathbf{Z}$ toward 1 and regularizing the relationships among channels across temporal offsets using

$$R_1 = \sum_{i,j} |1 - \mathbf{Z}[i,j]|, \quad R_2 = \frac{\sum_{i,j} \sum_t \mathbf{\Delta}[i,j,t]^2}{C^2},$$

respectively. The regularization term $R_2$ encourages consistency and stability in the learned inter-channel temporal relationships. Combining these loss terms results in the following training objective function to be applied across all data points $\mathcal{L} = \mathcal{L}_{\mathrm{ssl}} + \lambda_1 R_1 + \lambda_2 R_2$, where $\lambda_1$ and $\lambda_2$ control the extent and strictness of gating and temporal attention suppression.

## 3 EXPERIMENTS

In this section, we evaluate our model's embeddings on common downstream tasks, namely classification, forecasting, and anomaly detection, and benchmark our model against the current state-of-the-art models in each of the aforementioned downstream tasks. We expand on our datasets in Section H, and provide more details on downstream task training and baselines in Section J and Table 10 respectively.

**Forecasting**  We evaluate forecasting on the LSF benchmark suite (Wu et al. 2021, ETTh1/2, ETTm1/2, Weather) in the standard multi-horizon, multivariate setting with horizons 96/192/336/720. We compare against state-of-the-art foundation time-series models that are either (i) pre-trained for forecasting or (ii) pre-trained for representation learning and evaluated via dataset-specific linear probing (LP).

To produce point forecasts with CHARM, we consider three variants: (1) **CHARM+LP** — dataset-specific linear probes trained on frozen CHARM embeddings; (2) **CHARM + NLH** — a single, dataset-agnostic non-linear forecaster trained on frozen CHARM embeddings; and (3) **CHARM + NLH FT** — end-to-end training that back-propagates through CHARM to align the embeddings with the point-forecast objective. Per-dataset means (averaged across horizons) are reported in Table 1.

| Dataset | Toto | | Moirai_Small | | Moirai_Base | | Moirai_Large | | TimeMixer++ | | VisionTS | | MOMENT-LP | | PatchTST-LP | | CHARM + LP | | CHARM+NLH | | CHARM+NLH FT | |
|---|---|---|---|---|---|---|---|---|---|---|---|---|---|---|---|---|---|---|---|---|---|---|
| | MSE | MAE | MSE | MAE | MSE | MAE | MSE | MAE | MSE | MAE | MSE | MAE | MSE | MAE | MSE | MAE | MSE | MAE | MSE | MAE | MSE | MAE |
| Weather | 0.224 | **0.245** | 0.242 | 0.267 | 0.238 | 0.261 | 0.263 | 0.271 | 0.226 | 0.262 | 0.269 | 0.292 | 0.228 | 0.266 | 0.233 | 0.270 | 0.230 | 0.262 | 0.230 | 0.262 | **0.222** | 0.255 |
| ETTm1 | 0.396 | 0.378 | 0.448 | 0.410 | 0.382 | 0.404 | 0.390 | 0.389 | 0.368 | 0.378 | 0.373 | **0.371** | **0.344** | 0.379 | 0.350 | 0.382 | 0.413 | 0.432 | 0.416 | 0.437 | 0.411 | 0.428 |
| ETTm2 | 0.266 | 0.303 | 0.322 | 0.319 | 0.302 | **0.295** | 0.285 | 0.320 | 0.269 | 0.320 | 0.281 | 0.321 | 0.259 | 0.318 | 0.262 | 0.322 | 0.209 | 0.298 | 0.220 | 0.304 | **0.208** | 0.299 |
| ETTh1 | 0.435 | 0.413 | 0.400 | 0.423 | 0.432 | 0.440 | 0.510 | 0.469 | 0.395 | 0.419 | **0.392** | **0.405** | 0.418 | 0.422 | 0.428 | 0.438 | 0.554 | 0.532 | 0.555 | 0.557 | 0.557 | 0.526 |
| ETTh2 | 0.349 | **0.363** | 0.341 | 0.379 | 0.346 | 0.382 | 0.354 | 0.377 | 0.339 | 0.380 | 0.333 | 0.374 | 0.352 | 0.395 | 0.365 | 0.386 | 0.324 | 0.386 | 0.323 | 0.385 | **0.316** | 0.381 |

**Table 1:** Per-dataset mean MSE/MAE (lower is better). **Bold** = best, underline = second-best. The last three methods are grouped as **Ours**: CHARM-LP, *CHARM + Non-linear Head*, and *Fully Fine-tuned*.

Our *fully fine-tuned* model attains the lowest MSE on 3/5 datasets, outperforming significantly larger models trained on substantially bigger corpora (e.g., *Toto* and *Moirai*). By contrast, MAE leaders skew toward models optimized with MAE-aligned objectives (e.g., *Toto*, *VisionTS*, and *Morai*), which explains our relative MAE gap under an MSE-optimized head. See Section J.3 for forecasting-head details, training setup. Disaggregated results can be found in Table 23.

**Classification**  We evaluate our model on multivariate time series classification using the UEA dataset (Bagnall et al., 2018), benchmarking against semantic and reconstruction-based representation learning methods as well as specialized classification models. Results are summarized in Table 2. We test two protocols: (i) frozen encoder embeddings passed to an SVM with an `rbf` kernel, and (ii) a finetuned encoder with a linear classification head trained via cross-entropy loss.

| Method | Wins↑ | Avg. Acc.↑ | Total Correct↑ |
|---|---|---|---|
| TS2Vec | 2 | 78.1 | 7467 |
| T-Loss | 3 | 72.8 | 7141 |
| TS-TCC | 2 | 74.4 | 7118 |
| T-Rep | 3 | 78.5 | 7363 |
| MOMENT | 3 | 72.5 | 5414 |
| MiniROCKET | 4 | 77.6 | 7569 |
| CHARM$_{frozen+SVM}$ | 4 | 79.6 | 7431 |
| **CHARM$_{finetuned}$** | **5** | **80.9** | **7799** |

Table 2: Multivariate Classification Results.

Finetuning yields substantial gains over the frozen setting and baselines, achieving the highest number of wins, average accuracy, and unnormalized correct predictions. Against MiniRocket—a SOTA task-specific method—our frozen model is competitive despite MiniRocket's stronger raw scores (see Table 13), showing that our pretraining produces strong embeddings without task-specific adaptation. With finetuning, performance improves across all metrics, particularly in unnormalized scores (see Table 15), providing evidence that post-training alignment can substantially enhance downstream task performance.

**Anomaly Detection**  We use two tasks to evaluate our model's performance on anomaly detection. We use the Skoltech Anomaly Detection Benchmark (SKAB), to assess performance on real world multivariate datasets, as well as the popular UCR univariate anomaly detection dataset. For i) SKAB, we use baselines that consist of classical anomaly detection, CNN/LSTM based models, as well as more recent representation learning models. We reproduce these baselines by training a linear reconstruc-

| Method | F1↑ | FAR↓ | MAR↓ |
|---|---|---|---|
| MSCRED | 0.36 | 49.94 | 69.88 |
| T-Rep | 0.78 | **12.60** | 28.51 |
| MOMENT$_0$ | 0.79 | 14.20 | 26.98 |
| TS2Vec | 0.79 | 12.77 | 27.61 |
| MOMENT$_{LP}$ | 0.82 | 15.52 | 20.73 |
| **CHARM** | **0.86** | 19.35 | **12.69** |

Table 3: 34 SKAB Anomaly Detection Datasets

tion head on each of the 34 datasets in SKAB, and evaluating on the corresponding test instances. The evaluation setup in the SKAB test suite is uniformly applied to all models, which relies on using the errors at each time point in the test set to classify anomalies based on selecting an appropriate threshold (Section J.2). The setup for the reconstruction head (including optimization setup) is identical for all baselines for which the SKAB results were obtained by training[1]. As seen in Table 3, CHARM has the highest F1 score, followed closely by MOMENT, demonstrating strong performance on multivariate anomaly detection.

For (ii) UCR, we train our model with a reconstruction head on 46 UCR univariate anomaly detection datasets, which come from a diverse set of domains with varying types of anomalies. We benchmark our model against both task-specific anomaly detection methods and representation learning approaches, using the average adjusted F1 score as the standard evaluation metric. We report the per dataset scores, and wins in Table 16. Table 4 shows that CHARM has the highest average F1 score across all datasets.

| Method | F1↑ |
|---|---|
| Anomaly Transformer | 0.485 |
| DGHL | 0.415 |
| GPT4TS | 0.479 |
| TimesNet | 0.627 |
| MOMENT | 0.684 |
| **CHARM** | **0.754** |

Table 4: 46 UCR Univariate Datasets

### 3.1 ABLATIONS

To assess the impact of our architectural contributions, we conduct a series of ablation studies that isolate the benefits of (i) incorporating the proposed featurization layer based on temporal convolutional networks (TCNs) and (ii) modifying the text-based attention mechanism. Additional details on ablations experiments—covering description quality, choice of textual embedding model, comparison to naive text integrations, and the full experimental details—are reported in Appendix K.

---

[1]MOMENT$_0$ is used directly in reconstruction mode with no training, whereas MOMENT$_{LP}$'s reconstruction head is trained for each SKAB dataset, similar to other models.

| Category | Configuration | # Correct | Accuracy | ETTh1$_{T_f=168}$ MSE | ETTh1$_{T_f=168}$ MAE | ETTh2$_{T_f=168}$ MSE | ETTh2$_{T_f=168}$ MAE |
|---|---|---|---|---|---|---|---|
| Featurization Layer | w/ Patching | 4200 | 63% | 0.54 | **0.58** | 0.79 | 1.18 |
| | **TCN$_{no\ text}$** | **4713** | **68%** | 0.54 | 0.61 | **0.62** | 1.18 |
| TCN Variants | TCN$_{no\ text}$ | 4713 | 68% | 0.54 | 0.61 | 0.62 | 1.19 |
| | TCN$_{gate}$ | 4732 | 69% | 0.52 | 0.56 | 0.74 | 1.04 |
| | **TCN$_{conv}$** | **4897** | **71.4%** | **0.42** | **0.49** | **0.57** | **0.80** |
| Text Attention | $\varnothing$ (vanilla self-attn) | 4563 | 67.9% | 0.46 | 0.50 | 0.64 | 0.93 |
| | $\Delta$ | 4643 | 68.6% | 0.50 | 0.55 | 0.68 | 0.97 |
| | G | 4785 | 69.8% | 0.46 | 0.51 | 0.59 | 0.94 |
| | **$\Delta$ + G** | **4897** | **71.4%** | **0.42** | **0.49** | **0.57** | **0.80** |

**Table 5: Unified ablation study results.**[2] Top section compares TCN-based featurization to patching, middle section compares TCN variants, and bottom section evaluates different text attention mechanisms. We evaluate classification and forecasting metrics (`ETTh1`, `ETTh2`, horizon $T_f = 168$) (see Section K for full details).
**Featurization Layer:** w/Patching : using vanilla patch embedding layer; TCN$_{no\ text}$ = w/o text layers;
**TCN Variants:** TCN$_{no\ text}$ = w/o text layers; TCN$_{gate}$ = w/ text gating; TCN$_{conv}$ = w/ text-based convolutions.
**Text Attention:** G = text gating in attention layer; $\Delta$ = time-delta in attention layer.

Table 5 demonstrates that augmenting the vanilla time-series transformer architecture with our proposed text-based components yields consistent and substantial performance gains across both the featurization layers, as well as the attention layers.

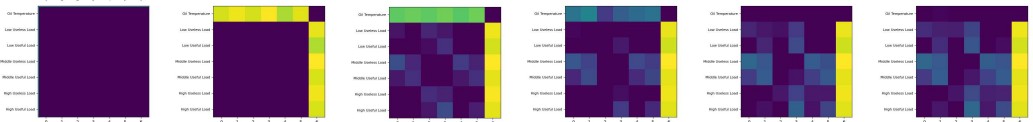

**Figure 6:** Evolution of Channel Gates for the `ETT` Dataset. A causal structure evolves over training, where the target causal variable `Oil Temperature` attends to all other independent channels but not vice versa. Extended discussion on evolution of channel gates can be found in Section L.2.

## 4 CONCLUSION

In this paper, we introduced `CHARM`, a foundation embedding model for multivariate time series that combines a description-aware temporal convolutional network with contextual attention over textual channel metadata. Using a JEPA-inspired self-supervised objective, `CHARM` learns enriched representations that go beyond reconstruction- or contrastive-based methods. Across diverse datasets and tasks, `CHARM` achieves competitive performance, with ablations confirming the importance of text featurization and attention layers. Furthermore, its heatmap visualizations (see Figure 6) provide interpretable insights into cross-channel dynamics.

As the first model to incorporate granular textual information into foundational time-series embeddings, `CHARM` opens promising avenues for deeper multimodal integration, multi-task architectures, and retrieval-augmented interpretive frameworks. At the same time, the model is constrained by its limited context length: it operates on the full-resolution input of length $T_{eff} = T \times C$, computing attention scores across all channel–time pairs. Future work will explore more efficient attention mechanisms to improve scalability to longer horizons and higher-dimensional inputs.

Finally, our results demonstrate that task-specific fine-tuning provides a noticeable lift in both forecasting and classification performance. This highlights the potential of systematic multi-task post-training strategies to further boost downstream performance and strengthen the role of foundation models in time-series analysis.

---

[2]**Featurization Layer and TCN** ablations uses $\Delta$+G; **Text Attention** ablations use TCN$_{conv}$

## 5 LLM USAGE STATEMENT

All authors used large language models to assist with text rephrasing, correction of grammatical errors, formatting, proof-reading for typos, and LaTeX typesetting.

## 6 REPRODUCIBILITY STATEMENT

We intend to release the pre-training dataset, including hand-annotated descriptions, along with performant infrastructure for dataset storage, loading, and preprocessing via GitHub. Since different datasets in our pipeline are subject to distinct usage licenses, we will additionally provide detailed guidelines for sourcing datasets that cannot be directly hosted on GitHub.

The architecture and hyperparameter configurations used for pretraining CHARM, as well as for downstream task-specific heads, are documented in the Appendix. We also include PyTorch-style pseudocode for the JEPA architecture (see Figure 1), together with efficient vectorized implementations of the text-attention layers (see Figure 9). We hope that this detailed documentation of architecture, hyperparameters, and pseudocode will enhance transparency and facilitate understanding of our model.

## 7 ETHICS STATEMENT

This work presents CHARM, a time series representation learning model developed to advance research on integrating textual information to enrich representation quality. As with other pretrained models, risks include the propagation of biases in training data and the environmental costs of compute-intensive training. To mitigate these concerns, we document data sources, model configurations, and training details to promote transparency. This research is intended for academic use and is not suitable for deployment in high-stakes decision-making contexts without additional safeguards.

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

## A   RELATED WORK

Historically, recurrent neural network architectures such as RNNs, LSTMs, and GRUs dominated time series modeling by capturing temporal dependencies through recursive hidden-state updates, achieving success across diverse tasks (Salinas et al., 2020). However, their sequential nature impeded parallelization, leading to slow training and difficulties in modeling long-range dependencies (Kalchbrenner et al., 2016; Pascanu et al., 2013; Zhou et al., 2021a; Kim et al., 2025).

With the emergence of Transformer architectures (Vaswani et al., 2017), significant advancements have occurred across multiple modalities, including images (Dosovitskiy et al., 2021), audio (Gong et al., 2021), video (Arnab et al., 2021), text (Devlin et al., 2018; Radford et al., 2018), and speech (Dong et al., 2018). Inspired by these successes, the time series community has adopted Transformer-based approaches, leading to notable innovations tailored specifically for temporal data (Zhou et al., 2021b; 2022; Ilbert et al., 2024; Wu et al., 2021; Zhang & Yan, 2023).

Simultaneously, self-supervised representation learning (SSRL), widely successful in domains such as vision and language, has demonstrated potential for extracting high-quality embeddings from vast amounts of unlabeled data. These embeddings facilitate downstream tasks—such as forecasting, classification, and anomaly detection—through lightweight task-specific heads. Analogous approaches have been adapted for time series, predominantly using contrastive self-supervised tasks (Yue et al., 2022; Fraikin et al., 2024; Franceschi et al., 2019; Tonekaboni et al., 2021). However, existing approaches typically produce models tailored to specific datasets, limiting their generalizability across arbitrary data sizes or channel configurations.

More recently, foundational models have revolutionized representation learning across natural language processing, computer vision, and audio (Devlin et al., 2019; Nussbaum et al., 2025; Assran et al., 2023; Kirillov et al., 2023; Baevski et al., 2020; Brown et al., 2020; Radford et al., 2021). In time series analysis, considerable progress has focused primarily on forecasting tasks (Das et al., 2024; Woo et al., 2024b; Ansari et al., 2024; Liu et al., 2024). Early foundational attempts predominantly addressed univariate series (Das et al., 2024; Ansari et al., 2024), though recent advancements have successfully extended to multivariate settings with sophisticated cross-channel modeling techniques (Woo et al., 2024b; Liu et al., 2024). Some more recent papers Cohen et al. (2025) have gone to great lengths to fully leverage the scaling laws observed in foundational time series models Edwards et al. (2025) in order to maximize their performance. Foundation embedding models specifically targeting time series representation learning have begun to emerge, leveraging reconstruction-based or next-step forecasting objectives (Goswami et al., 2024; Gao et al., 2024a; Trirat et al., 2024). However, these approaches either focus on univariate series or treat multivariate data as independent channels, inadequately capturing complex inter-channel dynamics. This substantially limits the representational richness and effectiveness of these models in realistic scenarios. See Table 6 for an overview of capabilities of key time series models.

**Joint-Embedding Predictive Architectures** have found notable success in visual domains by shifting the learning objective from pixel-level reconstruction to latent-space prediction. Extending this approach to video, Meta AI's Video JEPA with Variance–Covariance Regularization (VJ-VCR) (Drozdov et al., 2024) predicts future frame embeddings in a learned representation space while enforcing variance and covariance constraints to prevent collapse; this model outperforms generative reconstruction baselines on downstream tasks such as action recognition and video retrieval by capturing high-level spatiotemporal dynamics. Extensions such as MC-JEPA (Bardes et al., 2023) further demonstrate JEPA's flexibility by jointly learning motion (optical flow) and content features within a shared encoder–predictor framework, matching or surpassing unsupervised optical flow benchmarks and improving downstream segmentation tasks. In multimodal settings, TI-JEPA (Vo et al., 2025) integrates an energy-based JEPA with cross-modal encoders to align text and image embeddings, achieving superior results on multimodal sentiment analysis and visual question answering benchmarks by capturing complex semantic correspondences without reconstructing raw inputs. Complementing JEPA, bootstrapped embedding SSL methods like BYOL ("Bootstrap Your Own Latent") (Grill et al., 2020b) train an online network to predict the target network's representation of differently augmented views—updating the target via momentum averaging—and achieve strong results on ImageNet under linear evaluation without requiring negative pairs; this demonstrates that simple latent-space prediction objectives can match or exceed contrastive and reconstruction-based approaches in learning robust, generalizable representations. Together, these concrete instantiations

highlight JEPA's core advantage of filtering out low-level noise and focusing learning on high-level semantic structure, while bootstrapped SSL offers a practical, decoder-free paradigm for self-supervised representation learning, and motivate further exploration of these methods for time series.

**Multimodal text + time series models** Recent works have explored combining textual information with time series data through several novel approaches. For instance, Jin et al. (2024), Pan et al. (2024), and Sun et al. (2024) reprogram pretrained LLMs to handle time series input. These approaches fall under the *LLM-for-TS* or *TS-for-LLM* paradigms, where either an LLM is finetuned for time series data or the time series are transformed into token sequences consumable by an LLM. However, such methods do not directly leverage textual metadata; rather, they exploit the language modeling capabilities of models pretrained on large corpora of text. In contrast, Zhou et al. (2023) and Cai et al. (2023) explicitly incorporate textual information tied to data instances to improve time series representations, typically through contrastive objectives that align text and time series embeddings in a shared semantic space.

CHARM takes a fundamentally different approach to incorporating textual information. Instead of relying on instance-level labels to build contrastive training pairs, CHARM leverages sensor-level descriptions as metadata. These textual embeddings are integrated directly into the featurization stage (via TCNs) or used to augment the self-attention mechanism. Rather than aligning modalities, CHARM learns mappings from textual metadata to guide representation learning. This design enables CHARM to scale to massive datasets where instance-level text labels are unavailable or impractical, requiring only sensor descriptions to improve the quality of learned time series representations.

| Model | Multivariate | Channel Mixing | Equivariance | Foundational | Channel Aware |
|---|---|---|---|---|---|
| Tloss | ✓ | ✓ | ✗ | ✗ | ✗ |
| TS2Vec | ✓ | ✓ | ✗ | ✗ | ✗ |
| TNC | ✓ | ✓ | ✗ | ✗ | ✗ |
| Autoformer | ✓ | ✓ | ✗ | ✗ | ✗ |
| FEDformer | ✓ | ✓ | ✗ | ✗ | ✗ |
| PatchTST | ✓ | ✗ | ✗ | ✗ | ✗ |
| CrossFormer | ✓ | ✓ | ✗ | ✗ | ✗ |
| iTransformer | ✓ | ✓ | ✓ | ✓ | ✗ |
| UniTS | ✓ | ✓ | ✓ | ✓ | ✗ |
| TimesFM | ✗ | – | – | ✓ | ✗ |
| MOIRAI | ✓ | ✓ | ✓ | ✓ | ✗ |
| MOMENT | ✗ | – | ✓ | ✓ | ✗ |
| TREP | ✓ | ✓ | ✗ | ✗ | ✗ |
| TOTO | ✓ | ✓ | ✓ | ✓ | ✗ |
| TimeMixer++ | ✓ | ✓ | ✗ | ✗ | ✗ |
| CHARM | ✓ | ✓ | ✓ | ✓ | ✓ |

**Table 6:** a) **Multivariate:** Can handle multivariate data[1]
b) **Channel Mixing:** Architecture enables learnable cross-channel interactions[2]
c) **Equivariance:** Permuting the channels by a perturbation $P$ ensures the outputs are also identically permuted.
d) **Foundational:** Can flexibly accept data of any arbitrary number of channels or time window.
e) **Channel Aware:** Uses sensor information to learn better representations.

# B NOTATION

We denote matrices and tensors using boldface capital letters (e.g., $\mathbf{T}$, $\mathbf{E}$), and adopt *NumPy*-style indexing and slicing notation. Functions and operators are also denoted by bold capital letters, but are subscripted with $\theta$ to indicate parameterization, e.g., $\mathbf{E}_\theta$. The parameters $\theta$ may be learnable or fixed, depending on context. We reserve, $\mathbf{W}$ to represent the learnable weights in different layers of our architecture. An instance of a time series is represented as a tuple $\mathbf{t} = (\mathbf{T}, \mathbf{D}, \mathbf{pos})$. The first component, $\mathbf{T} \in \mathbb{R}^{T \times C}$, is a matrix of time series measurements, where $T$ denotes the number of time points and $C$ the number of channels. Each column $\mathbf{T}[:, i]$ corresponds to the uni-variate time series from channel $i$. The second component, $\mathbf{D}$, is an ordered list of length $C$, where each entry $\mathbf{D}[i]$ is a textual description of channel $i$, typically represented as a sentence or short passage. We assume that the descriptions in $\mathbf{D}$ are aligned with the corresponding columns of $\mathbf{T}$. The third component, $\mathbf{pos}$, represents the positional indices associated with the time series. We assume $\mathbf{pos} \in \mathbb{I}_+^T$ such

---

[1]Multivariate here simply refers to whether a model can ingest multiple input channels, i.e. whether it can feasibly operate on a $T \times C$ data input, where $C > 1$. This is independent of whether the model is able to learn channel interactions, which is explicitly outlined in the channel mixing column.

[2]We do not consider models that are fundamentally univariate, but perform late fusion of channels at the representation level (by pooling for example), to be capable of channel mixing.

that $|\mathbf{pos}| = T$. If not explicitly provided, we default to $\mathbf{pos} = [0, 1, \ldots, T-1]$. We denote the maximum time window size considered in our framework as $T_{\max}$.

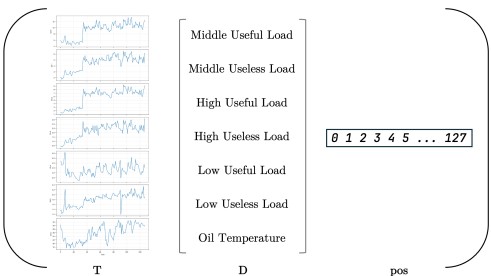

**Figure 7: Visualized representation of our data structure.** Note, that $\mathbf{T} \in \mathbb{R}^{T \times C}$, $|\mathbf{D}| = C$, and $\mathbf{pos} \in \mathbb{R}^T$

## C    IMPLEMENTATION DETAILS

We attempt to follow the general set of best practices developed in the field of self-supervised learning, specifically those applicable to the Self-Distillation (Balestriero et al., 2023) family of algorithms. We outline the key details here;

1. **Optimization Schedule** We use an AdamW optimizer to optimize our model. The learning rate follows a linear warmup followed by a cosine decay.

2. **Weight Initialization** We use a fixed $\mathcal{N}(0, 0.02)$ initialization which is commonly used in pretraining large transformer models (OLMo et al., 2025).

3. **Weight Decay Scheduling** We use a cosine schedule for increasing the optimizer's weight decay over the course of training which has been shown to be crucial for training stability.

4. **EMA Schedule for Target Encoder** We use an exponentially moving average with a momentum schedule that is increased gradually over the course of training.

The weight decay scheduling and EMA schedule are identical to IJEPA (Assran et al., 2023). Besides sweeping over a few learning rates, we perform no additional hyperparameter tuning on the rest of the hyperparameters due to limited compute, and list them in Table 7.

### C.1    ROTARY POSITION EMBEDDINGS

Rotary Position Embeddings (RoPE) (Su et al., 2024) differ from traditional additive positional embeddings in that they encode positional information by rotating the query and key vectors in a structured, position-dependent manner. Unlike fixed or learned additive embeddings, RoPE is applied at **each** layer of the self-attention computation, allowing the model to encode relative position information directly into the attention mechanism.

Let $\mathbf{Q}, \mathbf{K} \in \mathbb{R}^{B \times T \times D}$ denote the queries and keys, where $B$ is the batch size, $T$ is the sequence length, and $D$ is the hidden dimension. After linear projection and splitting into $H$ attention heads:

$$\mathbf{Q}_h, \mathbf{K}_h \in \mathbb{R}^{B \times T \times d}, \quad \text{with } d = D/H$$

RoPE applies a deterministic rotation to each head's query and key vectors. For each position $t$ and dimension index $i$, the rotation is defined as:

$$\text{RoPE}(\mathbf{x}_t)[2i] = \mathbf{x}_t[2i] \cos(\theta_{t,i}) + \mathbf{x}_t[2i + 1] \sin(\theta_{t,i}) \tag{1}$$

$$\text{RoPE}(\mathbf{x}_t)[2i + 1] = -\mathbf{x}_t[2i] \sin(\theta_{t,i}) + \mathbf{x}_t[2i + 1] \cos(\theta_{t,i}) \tag{2}$$

$$\theta_{t,i} = t \cdot \omega_i, \quad \omega_i = 10000^{-2i/d} \tag{3}$$

where $\mathbf{x}_t$ denotes the $t^{\text{th}}$ token's vector (query or key), and $\omega_i$ are predefined inverse frequency terms.

In our implementation, we operate on inputs of shape $\mathbf{X} \in \mathbb{R}^{B \times T \times C \times d}$, where $C$ represents the number of channels or sensors. To apply RoPE consistently across all channels, we broadcast the position encodings across the channel axis:

$$\widetilde{\mathbf{P}}_{b,t,c,:} = \mathbf{P}_{t,:}, \quad \forall\, b \in [1, B],\ c \in [1, C],\ t \in [1, T] \tag{4}$$

or equivalently, using broadcasting semantics:

$$\widetilde{\mathbf{P}} = \mathbf{P}[t, :] \longrightarrow \mathbb{R}^{B \times T \times C \times d} \tag{5}$$

This results in a broadcasted position encoding tensor $\widetilde{\mathbf{P}}$ where the same temporal position vector $\mathbf{P}_{t,:}$ is shared across all channels at time $t$, effectively associating the same position ID to multiple sensor tokens that occur at the same timestep.

## C.2 TEXT CONVOLUTION LAYER

### C.2.1 CONTRAST TO OTHER FEATURIZATION METHODS

Unlike typical TCNs, we concatenate activations across all intermediate layers to form a rich initial representation, see Figure 2. Our contextual TCN layer early in our model architecture closely relates to the concept of patching. Several recent foundation models for time series create non-overlapping static patches and project them through a single linear layer, e.g., (Das et al., 2024; Nie et al., 2023a; Woo et al., 2024b). These approaches can be generalized by interpreting convolution kernels as learnable linear mappings applied to strided segments of the data. Thus, our TCN layer represents a generalized, channel-aware extension of the patching concept.

### C.2.2 IMPLEMENTATION

To compute convolutions efficiently across all sensors and batches, we stack the convolutional kernels corresponding to each sensor description and reshape the input to treat the $B \times C \times H$ channels as independent time series. We then apply a grouped 1D convolution using `F.conv1d` with $B \times C \times H$ groups, where each element in the original $[B, T, C, H]$ input is treated as a separate time series along the time axis. This allows us to apply distinct filters for each batch, channel, and embedding dimension in parallel.

### C.2.3 INITIALIZATION

Despite the effectiveness of this mechanism, careful numerical stabilization of the convolution kernels is essential. To achieve this, we first apply a non-parametric LayerNorm to $z$-normalize the sensor embeddings, $\mathbf{E}_d$. The projection matrix within the kernel network is then initialized using Xavier normal initialization (Glorot & Bengio, 2010). Subsequently, we re-normalize the resulting kernels $\mathbf{W}_k$ as

$$\mathbf{W}_k = \mathbf{LayerNorm}(\mathbf{W}_k) \cdot \sqrt{\frac{2}{K}}$$

Since our TCN layer employs GeLU nonlinearities, this initialization approach aligns with Kaiming initialization principles, (He et al., 2015), and ensures stable activations, preventing them from progressively exploding or vanishing across convolution layers.

## C.3 MODEL SIZING

For the given hyperparameter set $N = 8, d = 128, \mathrm{ff}_{\text{dim}} = 4d$, our pretrained model is $\sim$7.1M parameters.

## C.4 ADDITIONAL MODIFICATIONS TO THE TRANSFORMER LAYERS

In line with recent developments in large scale pretraining of transformer based architectures, we implement several modifications that diverge from the original transformer architecture.

**SwiGLU** We replace the regular feedforward layers with a SwiGLU feedforward layer.

**QK-norm** We add a pre-attention layernorm to the queries and keys.

**Rotary Position Embeddings** Instead of using sinusoidal positional embeddings, we use rotary positional embeddings which are applied on the queries and keys at every layer. The positional indices are provided through the **pos** argument.

**Reordering Sublayers** We experiment with using 3 approaches to assess the optimal configuration of the transformer sublayers.

$$\begin{cases} x = \text{norm}(x + \text{SubLayer}(x)) & \text{Post Norm} \\ x = x + \text{SubLayer}(\text{norm}(x)) & \text{Pre Norm} \\ x = x + \text{norm}(\text{SubLayer}(x)) & \text{Swin Norm} \end{cases} \tag{6}$$

In the case of Pre Norm and Swin Norm, we also experiment with adding LayerNorms in the main transformer branch every $n$ layers, to ensure further stability.

## C.5 EFFICIENT COMPUTATION TECHNIQUES

### C.5.1 SLICE AND TILE ATTENTION LAYERS

To vectorize the process of generating the full $\bar{\mathbf{\Delta}}$ tensor, we provide the pytorch pseudocode versions of the naive and vectorized versions in Figure 8 and Figure 9.

**Figure 8:** Naïve attention-weight matrix construction

```python
def build_attention_weight_matrix(time_deltas: Tensor,
                                  T_proj: Tensor) -> Tensor:
    """
    Constructs the full attention weight matrix by explicit loops.
    Args:
        time_deltas: LongTensor, shape (T, T)
        T_proj:      Tensor, shape (B, C, C, 2*T - 1)
    Returns:
        attn:        Tensor, shape (B, C*T, C*T)
    """
    B, C, _, T1 = T_proj.shape
    T = time_deltas.size(0)
    assert 2 * T - 1 == T1

    attn = torch.zeros((B, C * T, C * T), device=T_proj.device)
    for i in range(T):
        for j in range(T):
            delta = time_deltas[i, j].item()
            block = T_proj[..., delta]    # (B, C, C)
            attn[..., i*C:(i+1)*C, j*C:(j+1)*C] = block
    return attn
```

# D JEPA

## D.1 DATASET GENERATION

The core principle of JEPA-based self-superived training involves producing representations for two augmented views originating from the same data instance. JEPA training aims to minimize a discrepancy measure (e.g., $\ell_1$ or $\ell_2$) between these representations. In vision, these views commonly result from image augmentations such as jittering, masking, or cropping.

Figure 10 presents a visual representation of our JEPA tasks, which rely on learning 1) causal representations and 2) smoothing representations.

**Figure 9:** Fast attention-weight matrix construction

```python
def build_attention_weight_matrix_fast(time_deltas: Tensor,
                                       T_proj: Tensor) -> Tensor:
    """
    Block-wise assembly via tensor indexing and reshape.
    """
    B, C, _, T1 = T_proj.shape
    T = time_deltas.size(0)
    assert 2 * T - 1 == T1

    # 1) Flatten index grid
    flat_idx = time_deltas.view(-1)          # shape (T*T,)

    # 2) Gather all needed projection slices at once
    gathered = T_proj.index_select(dim=-1, index=flat_idx)
    #     result: (B, C, C, T*T)

    # 3) Reshape to (B, C, C, T, T)
    gathered = gathered.view(B, C, C, T, T)

    # 4) Reorder to (B, T, C, T, C)
    gathered = gathered.permute(0, 3, 1, 4, 2)

    # 5) Collapse blocks into (B, C*T, C*T)
    return gathered.contiguous().view(B, C * T, C * T)
```

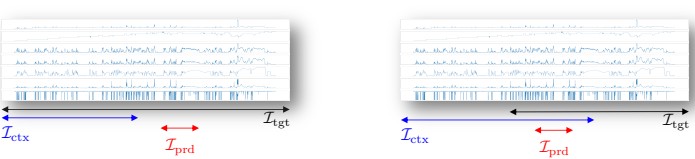

**Figure 10:** JEPA Tasks Visualized : Causal Prediction (left) Smoothing (right)

## D.2 JEPA ENCODERS – DEEP DIVE

In this section we dive a bit deeper into our implementation of the JEPA framework. We denote the TCN layer that featurizes our input time series as $\mathbf{F}$ and our encoder stack (of $N$ layers), as $\mathbf{E}$.

As outlined earlier, our featurizing layer converts a multivariate time series instance to an embedded version of the time series with the same leading dimensions, i.e.;

$$\mathbf{F} : \mathbb{R}^{T \times C} \to \mathbb{R}^{T \times C \times H}$$

On the other hand our encoder ingests the embedded time series and returns a contextually embedded time series while maintaining the same output dimensions i.e.;

$$\mathbf{E} : \mathbb{R}^{T \times C \times H} \to \mathbb{R}^{T \times C \times H}$$

Given this notation, our 3 JEPA networks (Context, Target, Predictor) can be represented as:

$$\mathbf{Context} \Rightarrow [\mathbf{F} \to \mathbf{E_1}] \tag{7}$$
$$\mathbf{Target} \Rightarrow [\mathbf{F} \to \mathbf{E_1}] \tag{8}$$
$$\mathbf{Predictor} \Rightarrow [\mathbf{DownProj} \to \mathbf{E_2} \to \mathbf{UpProj}] \tag{9}$$

Now, with this featurization and encoder layer stack, we provide a PyTorch style pseudocode of the JEPA framework, i.e. the data flow between the Context, Target, and Predictor encoders in Section D.2, Section D.2, and Section D.2.

| Category | Hyperparameter | Value |
|---|---|---|
| **Optimization Schedule** | Optimizer | AdamW |
| | $\epsilon$ | 1e-8 |
| | $\beta_1$ | 0.95 |
| | $\beta_2$ | 0.99 |
| | epochs | 100 |
| | gradient clipping | 2.0 |
| | $\lambda_1, \lambda_2$ | 1e-5 |
| | batch size [3] | 4 |
| | gradient accumulation | 2 |
| **Scheduler** | starting LR | 1e-8 |
| | final LR | 1e-6 |
| | starting weight decay | 0.04 |
| | final weight decay | 0.4 |
| | learning rate schedule | linear warmup $\rightarrow$ cosine decay |
| | weight decay schedule | cosine |
| | fraction of warmup epochs | 0.1 |
| | scale factor [4] | 1.25 |
| **Data** | window size | 512 |
| | stride | 128 |
| | minimum samples per dataset | 400 |
| | maximum samples per dataset | 1000 |
| **SSL Task Parameters** | number of targets | 4 |
| | $C_{\min}$ | 0.3 |
| | $C_{\max}$ | 0.4 |
| | $T_{\min}$ | 0.1 |
| | $T_{\max}$ | 0.2 |
| **JEPA Architecture** | encoder layers | 8 |
| | predictor layers | 4 |
| | encoder dim | 128 |
| | predictor dim | 64 |
| **Model Architecture** | feedforward layer | SwiGLU |
| | ff_dim_multiplier | 4 |
| | attention dropout | 0.01 |
| | norm | non-parametric layernorm |
| | attention configuration | pre-norm |

**Table 7:** Hyperparameters for full training pipeline

```python
class ContextTgtEncoder:
    def forward(self, x, ctx_idx):
        """
        x : [..., T, C]
        """
        x = self.featurizer(x)
        for layer in self.encoder_layers:
            x = layer(x, ctx_idx)
        return x
```

**Figure 11:** Context and Target Network

```python
class Predictor:
    def forward(self, ctx_embeds, ctx_idx, target_idx):
        """
        embeds : [..., T, C, H]
        target_pos : [..., T2]
        """
        x = self.downproj(ctx_embeds)  # [..., T, C, H1]
        mask_tokens = broadcast(self.mask_token, target_idx)  # [..., T2,
    C, H1]

        x = concat([x, mask_tokens])  # [..., T+T2, C, H1]
        pos = concat([ctx_idx, target_idx])

        for layer in self.encoder_layers:
            x = layer(x, pos)
        x = self.upproj(x)  # [..., T+T2, C, H]
        x = x[..., -target_idx.size(-2):, :, :]  # [..., T2, C, H]
        return x
```

**Figure 12:** Predictor Network

```python
class JEPA:
    def __init__(self):
        self.context_encoder = ContextTgtEncoder()
        self.target_encoder = copy_and_freeze_params(self.context_encoder
    )
        self.predictor = Predictor()

    def forward(self, x, ctx_idx, tgt_idx):
        """
        x : [..., T, C]
        """

        # get full embeddings
        full_embeds = self.target_encoder(x)

        # get context embeddings
        context_embeds = self.context_encoder(x[..., ctx_idx, :])

        # get predicted embeddings
        predicted_embeds = self.predictor(context_embeds, ctx_idx,
    tgt_idx)
        target_embeds = full_embeds[..., tgt_idx, :, :]

        # compute loss
        loss = loss_fn(predicted_embeds, target_embeds)
        return loss
```

**Figure 13:** JEPA

# E  HARDWARE

We use a cluster of 8 80GB NVIDIA A100 GPUs. We use Distributed Data Parallelism to speed up training, along with bf-16 mixed precision. Our models are implemented in PyTorch (Paszke et al., 2019), and training is done with PyTorch lightning (Falcon & The PyTorch Lightning team, 2019). We handle our configuration management using gin configs.

## F    LIMITATIONS

The primary limitations of our model are:

1. **Limited context lengths**
   Given our model's architecture, we are required to compute attention scores over the entire $C \times T$ input. As we do not rely on downsampling/patching, we compute the full $\mathcal{O}(C^2 T^2)$ attention matrix, which can be prohibitively large, especially for datasets with a large number of unrelated channels, or extremely long time horizons. Potential workarounds to this are computing the attention scores for only relevant channel pairs, based on pre-filtering similar channels based on the channel gating scores (high gating scores effectively clamp the attention scores completely, and self attention between these channels is effectively wasted compute). For large time horizons, a downsampling/patching layer can be appended to the encoder stack prior to the TCNs to operate on a lower effective time window.

2. **Access to sensor descriptions**
   As our model leverages channel descriptions directly in the featurization, and attention layers, we require access to high quality sensor descriptions that are provided with the dataset. Through our ablations conducted in Section K, we observe that noisy/arbitrary descriptions result in a moderate drop in model performance, thus highlighting the need for accompanying good enough quality descriptions. While such metadata is commonly available in practice, this requirement poses an overhead requirement for training and utilizing CHARM. For the UEA dataset, which provides detailed descriptions of each dataset in an accompanying document, we manually curated the sensor descriptions, which was a time consuming, and labor-intensive effort, and is not scalable to large unlabeled datasets.

## G    MODEL COMPLEXITY

Our pretraining was conducted on **8 A100 GPUs** over approximately **18 hours**, inclusive of minor overheads for dataset preprocessing, downstream evaluations, and logging. Training was performed with **bf16 mixed precision** under **distributed data parallelism**. A frozen text embedding model was invoked during training, served independently on a single L4 GPU. The **peak GPU memory usage** per device was **72.7 GB**, while **peak CPU utilization** remained at **4 GB**.

### G.1    ARCHITECTURAL CONTRIBUTIONS TO COMPLEXITY

Relative to conventional transformer architectures, the primary increase in model complexity arises from the inclusion of the **temporal convolutional (TCN) layer** and **text-attention layers**. For a representative configuration—where the text embedding dimension is 384, the time-series embedding dimension is 256, and the convolution kernel size is 8—the parameter counts for the additional modules are enumerated in Table 8.

| Layer | Count | Number of Parameters |
|---|---|---|
| TCN | $\mathbf{D}_{\text{text}} \times H \times K$ | $384 \times 256 \times 8 = 786{,}432$ |
| Gating Layer | $H^2$ | $256^2 = 65{,}536$ |
| Time-Delta Layer | $2 \times \mathbf{D}_{\text{text}} \times T_{\max}$ | $2 \times 384 \times 1500 = 1{,}152{,}000$ |

**Table 8:** Parameter counts for the TCN and text-attention modules.

In total, these components introduce approximately **2 million additional parameters**, corresponding to ~25% of the overall model size.

### ENCODER COMPOSITION

The pretraining framework employs **three modules**: a **context encoder**, a **target encoder**, and a **predictor**. The target encoder parameters are non-trainable and are tied to the context encoder parameters but consume GPU memory equivalent to the context encoder during forward passes. The predictor is comparatively lightweight, operating at lower dimensionality with fewer layers.

The **parameter distribution** across modules is as follows:

- Context encoder: **7.1M parameters**
- Target encoder: **7.1M parameters (frozen)**
- Predictor: ∼**4M parameters**

The combined model therefore comprises ∼**18.2M parameters**, of which ∼**11.1M are trainable**. For computing the embeddings at inference we only utilize the context encoder, which means the embeddings are the output of a **7.1M** parameters model.

## H    DATASETS

Here we provide a list of dataset sources we used to train our model. Wherever sensor names were not readily available, we manually curate the sensor descriptions from the dataset specifications.

**UEA Dataset**    The UEA Dataset is a popular publicly available dataset used for benchmarking time series classification algorithms. We restrict ourselves to a subset of the full 30 datasets, as not all of them have meaningful sensor descriptions. For a few of the datasets within UEA, we manually annotate the descriptions based on the official paper (Bagnall et al., 2018).

**Liu-ICE Machine Fault Dataset**    The Liu-ICE Machine Fault Dataset is a real world fault diagnosis dataset which consists of data collected from an internal combustion engine test bench. The dataset consists of multiple different kinds of fault scenarios, and comes with a publicly available benchmark.

**Electricity Transformer Dataset**    The Electricity Transformer Dataset (ETTDataset/ETDataset) is a widely used dataset for time series forecasting, which contains data of dynamic power loads in an electric power grid located in China. This dataset contains 4 sub-datasets (`ETTh1`, `ETTh2`, `ETTm1`, `ETTm2`), which operate at different granularities.

**Weather**    The Weather Dataset from the MPI is a real world dataset of meteorological indicators for the year of 2020.

**Electricity**    The Electricity dataset contains hourly consumption from multiple consumers from 2012 to 2014.

**Illness**    The Illness Dataset includes weekly records for patients suffering from influenza like illnesses collected by the CDC.

**SKAB - Skoltech Anomaly Benchmark Dataset**    The SKAB dataset is designed for evaluating anomaly detection, targeted at two main problems : outlier detection and changepoint detection

**Gas Sensor Array Modulation**    The Gas Sensor Array Modulation from the UCI Machine Learning Repository is collection of time-series recordings obtained from an array of metal-oxide gas sensors.

**Machinery Fault Dataset**    The Machinery Fault Dataset comprises six different simulated states: normal function, imbalance fault, horizontal and vertical misalignment faults and, inner and outer bearing faults from a machinery fault simulator.

**Metro PT-3 Dataset**    The MetroPT-3 dataset is a multivariate time series collection created for predictive maintenance in the railway industry. It consists of over 1.5 million records (instances) captured at 1Hz from a train compressor's Air Production Unit (APU) over the period from February to August 2020.

**Unleashing the Power of Wearables**    The Human Activity Recognition Trondheim (HEART) dataset is a professionally annotated collection designed for developing machine learning algorithms capable of recognizing human activities in a free-living environment. Created at the Norwegian University of Science and Technology (NTNU), it features 22 subjects who wore two 3-axis Axivity AX3 accelerometers for approximately 2 hours each while performing various daily tasks. The sensors

were placed on the right thigh and lower back, providing multivariate time series data sampled at 50Hz.

**Predictive Maintenance of Hydraulic Systems**    The Predictive Maintenance of Hydraulic Systems dataset contains multivariate time series data collected from a hydraulic test rig. This dataset includes sensor readings—such as pressures, volume flows, temperatures, and more—recorded during load cycles of the hydraulic system.

We provide a summary of the specifications of each dataset in Table 9. If a dataset is present in a downstream benchmark, we only include the defined "train" subset of the full dataset, to prevent the model from optimizing an SSL loss over the test dataset samples.

| Dataset Name | #Timestamps | #Channels |
|---|---:|---:|
| **Open-Source/Kaggle Datasets** | | |
| Appliances Energy Prediction | 19,735 | 26 |
| Gas Sensor Array Temperature Modulation | 3,843,160 | 19 |
| Household Electric Power Consumption | 2,075,259 | 7 |
| Machinery Fault Diagnosis | 487,748,049 | 8 |
| MetroPT-3 Dataset | 1,516,948 | 15 |
| Predictive Maintenance of Hydraulics System | 132,300 | 17 |
| SKAB - Skoltech Anomaly Benchmark | 46,860 | 8 |
| Unleashing the Power of Wearables | 6,461,328 | 6 |
| Liu | 288,623 | 10 |
| **UEA Datasets** [5] | | |
| NATOPS | 9180 | 24 |
| Epilepsy | 28222 | 3 |
| Articulary Word Recognition | 39600 | 9 |
| UWave Gesture Library | 37800 | 3 |
| Cricket | 129276 | 6 |
| ERing | 1950 | 4 |
| Character Trajectories | 169218 | 3 |
| Finger Movements | 15800 | 28 |
| SelfRegulation SCP1 | 240128 | 6 |
| Basic Motions | 4000 | 6 |
| Atrial Fibrillation | 9600 | 2 |
| Hand Movement Direction | 64000 | 10 |
| Handwriting | 22800 | 3 |
| Libras | 8100 | 2 |
| LSST | 88,524 | 6 |
| Racket Sports | 4530 | 6 |
| **Forecasting Benchmark Datasets** | | |
| ETTh1 | 17,420 | 7 |
| ETTh2 | 17,420 | 7 |
| ETTm1 | 69,680 | 7 |
| ETTm2 | 69,680 | 7 |
| Weather | 52,696 | 21 |
| Illness | 966 | 7 |

**Table 9:** Overview of datasets categorized into Open-Source/Kaggle, UEA, and Forecasting benchmark datasets.

---

[5]The UEA Datasets are provided as windowed instances, i.e. they are not hosted as contiguous, chronological blocks of shape $T \times C$, but rather stored as $N \times T' \times C$. Here, we compute the "# of timesteps" as $N \times T'$, although there may be redundant overlaps based on how the data was collected and labelled.

# I DATA LOADING

To enable efficient data loading, we perform under/over sampling to balance the datasets. The degree of under/over sampling is controlled by the $t_1$ : `min_samples_per_dataset` and $t_2$ : `max_samples_per_dataset` parameters, which upsamples or downsamples the data if the number of samples is either $< t_1$ or $> t_2$ respectively.

Following this, each dataset is handled by its own dataloader, which cyclically yields batches of data from each dataset at every training step. This is handled internally by `pytorch lightning`'s `CombinedLoader` method, which yields a batch from each dataloader (if the iterator is not yet exhausted). As a result, our `effective batch size` [6] per step is now computed as :

$$(\text{batch size}) \times (\text{\# of GPUs}) \times (\text{\# of datasets}) \times (\text{grad\_accum\_steps}) \tag{10}$$

At the beginning of every epoch, we reload all datasets, which results in fresh under/over sampling indices. This enables the support of larger datasets to be incrementally covered over multiple epochs of training.

The JEPA tasks are randomly sampled after the datasets are sampled, which results in fresh context and target masks for repeated samples. This avoids the exact same sample being repeated several times in an epoch for underrepresented datasets, due to stochasticity in how the masks are generated.

## I.1 PERTURBATIONS

Our augmentation design is directly motivated by failure modes frequently observed in real-world time-series data—particularly in industrial and sensor-driven applications— where channel- or block-level gaps occur due to intermittent sensor outages, network disruptions, or scheduled maintenance. To build robustness against such artifacts, we incorporate two principled time-domain masking strategies:

- **Uniform segment masking:** masks a contiguous temporal segment across all channels, simulating system-level events such as edge-cache dropout or network-wide packet loss.
- **Channel-selective masking:** applies the same temporal mask to a randomly selected subset of channels, capturing sensor-specific anomalies such as probe failure or drifting instrumentation.

These perturbations are applied solely to the context encoder's input during training, while the teacher view remains unperturbed. This asymmetry forces the model to leverage broader temporal and cross-channel structure for representation learning, in line with the JEPA framework's core principle of predicting masked target representations rather than raw values.

The augmentation functions are tailored to the time-series domain but echo proven techniques in analogous modalities. For instance, our segment masking is a temporal analogue to SpecAugment's `TimeMasking`, a canonical augmentation for large-scale speech models (implemented in `torchaudio.transforms.TimeMasking`). Similar masking-based augmentations have also been adopted in recent time-series representation learning methods such as TEST (Sun et al., 2024).

Hyperparameters controlling mask width and frequency were chosen based on prior experience with industrial time-series systems. Due to computational budget constraints, we did not conduct a systematic hyperparameter sweep, instead prioritizing augmentations with clear interpretability and real-world grounding.

| Tasks | Supervision | Datasets | # datasets | Metrics | Baselines |
|---|---|---|---|---|---|
| Classification | Frozen SVM; Finetuned+Linear | UEA | 21 | Accuracy, Wins, Total Correct | MiniROCKET, TS2Vec, T-Loss, TS-TCC, T-Rep, MOMENT |
| Anomaly Detection | Frozen reconstructor (linear head) | UCR-AD | 46 | Adjusted Best F1 | Anomaly Transformer, DGHL, GPT4TS, TimesNet, MOMENT (0, LP), TS2Vec, T-Rep |
| Anomaly Detection | Frozen reconstructor (linear head) | SKAB | 34 | F1, FAR, MAR | $T^2$, $T^2+Q$ (PCA), Isolation Forest, MSET, Feed-Forward AE, Conv-AE, LSTM-AE, VAE, LSTM-VAE, MSCRED, TS2Vec, T-Rep, MOMENT |
| Long-horizon Forecasting | Per-dataset linear probe (frozen); universal non-linear head (frozen encoder); universal non-linear head (unfrozen encoder) | ETTh1 ETTh2 ETTm1 ETTm2 Weather Exchange Rate Illness | 7 | MSE, MAE | T-Rep, TS2Vec, PatchTST, MOMENT, Toto, TIMEMIXER++, Moirai, VisionTS |

**Table 10:** Unified baseline summary by task, now including dataset counts.

## J  EXPERIMENTS

### J.1  CLASSIFICATION

#### J.1.1  UEA CLASSIFICATION BENCHMARK

**Dataset**    We evaluate our model on the popular UEA Dataset which serves as a standard time series classification benchmark for multivariate data. We consider a subset of 21 UEA datasets (list in Table 11) that cover a diverse set of tasks and domains. We select these datasets on the basis of what our model's default context length can handle in a single GPU. As we modify the attention mechanism directly, we cannot leverage existing efficient implementations, and thus are restricted by a maximum context window size. Formally, we select the subsets based on the following rule: num channels $< 50$, num timestamps $< 1500$.

**Task Description**    Given a labeled time series data instance $(X, y)$, where $X$ is a multivariate time series, and $y$ corresponds to a supervised label corresponding to $X$, our goal is to learn a classifier $h$ to minimize test error, i.e. $\epsilon = \mathbb{E}_{(x,y)\sim\mathbb{P}_{(x,y)}}[\mathbb{1}(h(x) \neq y)]$.

**Downstream Setup**    We evaluate the quality of our representations, $Z \in \mathbb{R}^{B\times T\times C\times H}$ in the following setups.

1. **Frozen + off-the-shelf non-linear classifier (SVM)**
   Similar to Goswami et al. (2024), we flatten our embeddings, $Z$ and feed them to an SVM with the standard set of hyperparameters proposed in Franceschi et al. (2019), which are also used in `T-Rep`, `TS2Vec`, `T-Loss`, etc. The hyperparameters are chosen for each dataset separately, using 5-fold cross validation on the train set.

2. **Finetuned + linear probe**
   Similar to the linear probing setup in Goswami et al. (2024), we finetune the encoder for each dataset separately. We flatten the embeddings $Z$, and feed them to a single linear layer which maps the embeddings to a vector of logits, trained with a cross entropy loss. The training hyperparameters are listed in Table 12, which is the same for all UEA datasets.

---

[6]The "# of datasets" technically refers to the number of unexhausted datasets on that training step, as each dataset has a different number of samples.

| Dataset | Channels | Length | Included |
|---|---|---|---|
| ArticularyWordRecognition | 9 | 144 | ✓ |
| AtrialFibrillation | 2 | 640 | ✓ |
| BasicMotions | 6 | 100 | ✓ |
| CharacterTrajectories | 3 | 182 | ✓ |
| Cricket | 6 | 1 197 | ✓ |
| Epilepsy | 3 | 206 | ✓ |
| ERing | 4 | 65 | ✓ |
| FingerMovements | 28 | 50 | ✓ |
| HandMovementDirection | 10 | 400 | ✓ |
| Handwriting | 3 | 152 | ✓ |
| JapaneseVowels | 12 | 29 | ✓ |
| Libras | 2 | 45 | ✓ |
| LSST | 6 | 36 | ✓ |
| NATOPS | 24 | 51 | ✓ |
| PenDigits | 2 | 8 | ✓ |
| Phoneme | 11 | 217 | ✓ |
| RacketSports | 6 | 30 | ✓ |
| SelfRegulationSCP1 | 6 | 896 | ✓ |
| SelfRegulationSCP2 | 7 | 1 152 | ✓ |
| SpokenArabicDigits | 13 | 93 | ✓ |
| UWaveGestureLibrary | 3 | 315 | ✓ |
| DuckDuckGeese | 1 345 | 270 | × |
| EigenWorms | 6 | 17 984 | × |
| EthanolConcentration | 3 | 1 751 | × |
| FaceDetection | 144 | 62 | × |
| Heartbeat | 61 | 405 | × |
| InsectWingbeat | 200 | 78 | × |
| MotorImagery | 64 | 3 000 | × |
| PEMS-SF | 963 | 144 | × |
| StandWalkJump | 4 | 2 500 | × |

**Table 11:** An overview of the subset of UAE data sets included in the evaluation of CHARM.

| Hyperparameter | Value |
|---|---|
| Batch size | 16 |
| Learning rate | $1e-4$ |
| Weight decay | $1e-4$ |
| Epochs | 500 |
| Optimizer | Adam |
| Label smoothing | 0 |

**Table 12:** Training hyperparameters for finetuning setup

**Baselines**    To position ourselves in the existing landscape of time series classification methods, we include baselines from the following set of approaches:

1. Time Series Classification Models: `MiniRocket`

2. Semantic Representation Learning Models : `T-Rep`, `TS2Vec`, `T-Loss`, `TS-TCC` etc.

3. Reconstruction Based Representation Learning Models : `MOMENT`

Given our limited compute availability, all baseline results reported in the results table are drawn from prior published work. We restrict our comparison to models with results on the majority of the UEA datasets, and therefore exclude models with incomplete or missing UEA coverage (e.g., `UniTS`).

**Metrics**   To compare our model's performance on the combined set of UEA Datasets, we measure 3 quantities:

**Average Accuracy.**   For dataset $i$ with $n_i$ samples:

$$\text{Acc}_i = \frac{1}{n_i} \sum_{j=1}^{n_i} \mathbf{1}[h(x_{ij}) = y_{ij}],$$

and the average accuracy across $D$ datasets is

$$\text{AvgAcc} = \frac{1}{D} \sum_{i=1}^{D} \left( \frac{1}{n_i} \sum_{j=1}^{n_i} \mathbf{1}[h(x_{ij}) = y_{ij}] \right).$$

**Number of Correctly Classified Samples.**

$$\text{NumCorrect} = \sum_{i=1}^{D} \sum_{j=1}^{n_i} \mathbf{1}[h(x_{ij}) = y_{ij}].$$

**Number of Wins.**   For $M$ models $\{h_m\}_{m=1}^{M}$, define accuracy of model $m$ on dataset $i$ as

$$\text{Acc}_{i,m} = \frac{1}{n_i} \sum_{j=1}^{n_i} \mathbf{1}[h_m(x_{ij}) = y_{ij}].$$

The number of wins for model $m$ is

$$\text{Wins}(m) = \sum_{i=1}^{D} \mathbf{1}\left[ \text{Acc}_{i,m} = \max_{m'} \text{Acc}_{i,m'} \right].$$

We report average accuracy and number of wins as they are standard measures used in other papers that use the UEA benchmark, however, as noted in Fleming & Wallace (1986), we would like to highlight that relying on averages of arithmetic means in such setups might be misleading, as the number of test samples vary significantly per dataset (see Figure 14). As a result we additionally include unnormalized scores, which we empirically observe to be a relatively less noisier metric to track during training.

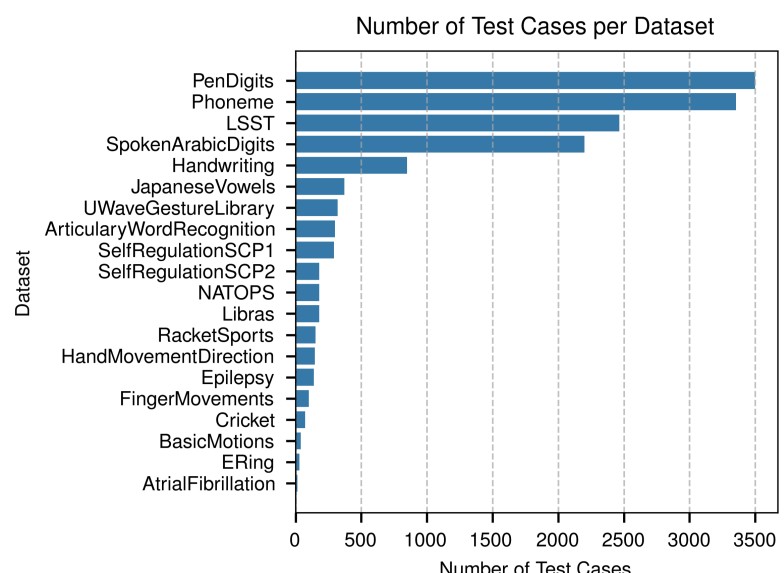

**Figure 14:** Sizes of different UEA Datasets

**Results**  Results for CHARM under the frozen + SVM setup are presented in Table 13, while the finetuned version is reported in Table 15. Overall, CHARM demonstrates strong aggregate performance, particularly in terms of average accuracy and total correct predictions across datasets. Moreover, we observe competitive results on datasets excluded from pre-training (`JapaneseVowels`, `PhonemeSpectra`, `PenDigits`), highlighting the strong generalization ability of the learned embeddings. The substantial improvement from finetuning on individual datasets suggests that post-training strategies can be effectively used to adapt the model for classification tasks.

| Dataset | TS2Vec | T-Loss | TS-TCC | T-Rep | MOMENT | MiniROCKET | CHARM$_{frozen + SVM}$ |
|---|---|---|---|---|---|---|---|
| AtrialFibrillation | 29 | 13 | 27 | 35 | 20 | 20 | 47 |
| Articulary/WordRecognition | 97 | 94 | 95 | 97 | 99 | 98 | 99 |
| BasicMotions | 100 | 100 | 100 | 100 | 100 | 100 | 98 |
| CharacterTrajectories | 99 | 99 | 99 | 99 | — | 99 | 98 |
| Cricket | 95 | 97 | 92 | 96 | 99 | 97 | 96 |
| ERing | 90 | 13 | 90 | 94 | 96 | 93 | 96 |
| Epilepsy | 96 | 97 | 96 | 97 | 99 | 100 | 98 |
| FingerMovements | 50 | 58 | 46 | 50 | 49 | 42 | 59 |
| HandMovementDirection | 42 | 35 | 24 | 54 | 32 | 41 | 54 |
| Handwriting | 46 | 45 | 50 | 41 | 31 | 24 | 33 |
| LSST | 56 | 51 | 47 | 53 | 41 | 67 | 60 |
| Libras | 86 | 88 | 82 | 83 | 85 | 94 | 83 |
| NATOPS | 90 | 92 | 82 | 80 | 83 | 92 | 82 |
| RacketSports | 89 | 86 | 82 | 88 | 80 | 88 | 86 |
| SelfRegulationSCP1 | 79 | 84 | 82 | 82 | 84 | 88 | 82 |
| UWaveGestureLibrary | 88 | 88 | 75 | 89 | 91 | 91 | 91 |
| SpokenArabicDigits | 99 | 91 | 97 | 99 | 98 | 99 | 97 |
| SelfRegulationSCP2 | 55 | 54 | 53 | 59 | 48 | 49 | 58 |
| Japanese Vowels † | 97 | 99 | 93 | 96 | 72 | 92 | 97 |
| Phoneme Spectra † | 24 | 22 | 25 | 23 | 23 | 28 | 20 |
| Pen Digits † | 98 | 98 | 97 | 97 | 97 | 97 | 98 |
| **Wins** | 2 | 5 | 2 | 3 | 3 | **7** | 4 |
| **Avg. Accuracy** | 76.4 | 71.6 | 73.1 | 76.8 | 71.3* | 76.2 | **77.6** |
| **Total Correct** | 7171 | 6836 | 6835 | 7075 | 5204* | **7284** | 7139 |

**Table 13:** Comparison of classification accuracy across multiple datasets and models. Datasets marked with † were not included in pre-training CHARM. *MOMENT does not report scores for `CharacterTrajectories`, and we exclude it while calculating MOMENT's scores.

| Hyperparameter | Value |
|---|---|
| C | {0.0001, 0.001, 0.01, 0.1, 1, 10, 100, 1000, 10000} |
| kernel | {'rbf'} |
| degree | {3} |
| gamma | {'scale'} |
| coef0 | {0} |
| shrinking | {True} |
| probability | {False} |
| tol | {0.001} |
| cache_size | {200} |
| class_weight | {None} |
| verbose | {False} |
| max_iter | {10000000} |
| decision_function_shape | {'ovr'} |
| random_state | {None} |

**Table 14:** SVM Hyperparameter Grid

| Dataset | TS2Vec | T-Loss | TS-TCC | T-Rep | MOMENT | MiniROCKET | CHARM$_{frozen+SVM}$ | CHARM$_{finetune}$ |
|---|---|---|---|---|---|---|---|---|
| AtrialFibrillation | 29 | 13 | 27 | 35 | 20 | 20 | **47** | 40 |
| Articular/WordRecognition | 97 | 94 | 95 | 97 | **99** | 98 | 99 | **99** |
| BasicMotions | **100** | **100** | **100** | **100** | **100** | **100** | 98 | **100** |
| CharacterTrajectories | 99 | **99** | 99 | 99 | – | 99 | 98 | 99 |
| Cricket | 95 | 97 | 92 | 96 | **99** | 97 | 96 | 94 |
| ERing | 90 | 13 | 90 | 94 | 96 | 93 | **96** | 94 |
| Epilepsy | 96 | 97 | 96 | 97 | 99 | **100** | 98 | 99 |
| FingerMovements | 50 | 58 | 46 | 50 | 49 | 42 | **59** | 57 |
| HandMovementDirection | 42 | 35 | 24 | 54 | 32 | 40 | **54** | 51 |
| Handwriting | 46 | 45 | **50** | 41 | 31 | 24 | 33 | 36 |
| LSST | 56 | 51 | 47 | 53 | 41 | 67 | 60 | **71** |
| Libras | 86 | 88 | 82 | 83 | 85 | **94** | 83 | 87 |
| NATOPS | 90 | 92 | 82 | 80 | 83 | 92 | 82 | **92** |
| RacketSports | **89** | 86 | 82 | 88 | 80 | 88 | 86 | 86 |
| SelfRegulationSCP1 | 79 | 84 | 82 | 82 | 84 | 88 | 82 | **91** |
| UWaveGestureLibrary | 88 | 88 | 75 | 89 | 91 | **91** | 91 | 88 |
| SpokenArabicDigits | 99 | 91 | 97 | **99** | 98 | 99 | 97 | 98 |
| SelfRegulationSCP2 | 55 | 54 | 53 | **59** | 48 | 49 | 58 | 57 |
| Japanese Vowels | 97 | 99 | 93 | 96 | 72 | 92 | 97 | 98 |
| **Wins** | 2 | 3 | 2 | 3 | 3 | 4 | 4 | **5** |
| **Avg. Accuracy** | 78.1 | 72.8 | 74.4 | 78.5 | 72.5* | 77.6 | 79.6 | **80.9** |
| **Total Correct** | 7467 | 7141 | 7118 | 7363 | 5414* | 7569 | 7431 | **7799** |

**Table 15:** Performance comparison across datasets in %. Best results per dataset are boldfaced, and the best count is reflected in the win statistics. PenDigits and PhonemeSpectra are omitted from the finetuned comparisons, due to the size of these datasets, and the associated training compute and time required. *MOMENT does not report scores for CharacterTrajectories, and we exclude it while calculating MOMENT's scores.

## J.2 ANOMALY DETECTION

### J.2.1 UCR ANOMALY DETECTION BENCHMARK

**Dataset** The UCR anomaly detection dataset Dau et al. (2018) is a popular open-source univariate anomaly detection dataset. The dataset consists of >100 datasets from varying domains. We restrict ourselves to the same subset of 46 datasets used in MOMENT (Goswami et al., 2024) which cover a diverse set of sources.

**Task Description** Each dataset in the UCR archive is provided with a "clean" train split, and a corresponding test split. The standard setup in this task involves training a model to reconstruct clean samples (i.e. with no anomalies), and then use this model on the test set to reconstruct the data. The mean squared error is computed in a point-wise sense on all timestamps in the test set. If the error corresponding to each timestamp exceeds a certain threshold, we classify that timestamp as anomalous. For a fair comparison, we use the same sweep over the error thresholds as used in MOMENT, which uses 100 samples on a linearly spaced grid from the lowest error to the highest error in the test set errors across all timestamps. Then, we compute an adjusted F1 score, which is standard practice in benchmarking anomaly detection models, for each threshold, and report the best adjusted

F1 score for each dataset. For this experiment, the embedding model is frozen and only the linear reconstruction head is trained.

**Downstream Setup** Our model is extended with a reconstruction head, which consists of a linear layer that maps embeddings back to the raw time series values, i.e. $Z \in \mathbb{R}^{T' \times 1 \times H} \to Z_t \in \mathbb{R}^T$. We empirically observe better results by applying an `AvgPool` on the embeddings (with a stride of 8) before the reconstruction head, as it potentially reduces the high fidelity of our per time point embeddings. Consequently, the linear layer is then of dimensions $\mathbb{R}^{H \times 8}$.

**Baselines** To ensure a fair comparison with models from varying classes, i.e. task-specific vs general representation learning, we benchmark ourselves against the same set of models used in `MOMENT`, which consist of state-of-the-art anomaly detection models, as well as general representation learning methods. These consist of: `Anomaly Detection Transformer`, `DGHL`, `GPT4TS`, `TimesNet` and `MOMENT`. Given our limited compute availability, all baseline results reported in the results table are drawn from prior published work and we limited ourselves to reported models in the `MOMENT` paper. We exclude `T-Rep` and `TS2Vec`, as they do not report results on this dataset/task.

**Metrics** We evaluate performance by measuring an adjusted F1 score for each dataset after optimizing the threshold for each dataset separately.

**Results** We report adjusted F1 scores across all datasets and models in Table 16. While model performance varies considerably by dataset, CHARM achieves strong overall results in terms of both average F1 score and total wins.

| Dataset | Anomaly Transformer | MOMENT | CHARM | DGHL | GPT4TS | TimesNet |
|---|---|---|---|---|---|---|
| 1sddb40 | 0.03 | 0.54 | 0.99 | 0.39 | 0.19 | 0.68 |
| BIDMC1 | 0.99 | 1.00 | 1.00 | 1.00 | 1.00 | 1.00 |
| CHARISfive | 0.01 | 0.13 | 1.00 | 0.02 | 0.02 | 0.08 |
| CHARISten | 0.02 | 0.11 | 0.12 | 0.01 | 0.01 | 0.03 |
| CIMIS44AirTemperature3 | 0.06 | 0.98 | 0.98 | 0.50 | 0.18 | 0.47 |
| CIMIS44AirTemperature5 | 0.39 | 0.99 | 0.85 | 0.96 | 0.20 | 0.71 |
| ECG2 | 1.00 | 1.00 | 1.00 | 0.62 | 0.90 | 1.00 |
| ECG3 | 0.36 | 0.98 | 0.93 | 0.80 | 0.84 | 0.48 |
| Fantasia | 0.75 | 0.95 | 0.97 | 0.66 | 0.87 | 0.55 |
| GP711MarkerLFM5z4 | 0.93 | 1.00 | 0.64 | 0.50 | 0.64 | 0.95 |
| GP711MarkerLFM5z5 | 0.76 | 0.97 | 0.75 | 0.31 | 0.48 | 0.90 |
| InternalBleeding5 | 0.94 | 1.00 | 1.00 | 1.00 | 0.92 | 1.00 |
| Italianpowerdemand | 0.01 | 0.74 | 0.17 | 0.59 | 0.01 | 0.44 |
| Lab2Cmac011215EPG5 | 0.99 | 0.98 | 1.00 | 0.34 | 0.60 | 0.99 |
| Lab2Cmac011215EPG6 | 0.41 | 0.10 | 0.12 | 0.26 | 0.10 | 0.17 |
| MesoplodonDensirostris | 1.00 | 0.84 | 1.00 | 0.79 | 1.00 | 1.00 |
| PowerDemand1 | 0.87 | 0.44 | 0.43 | 0.49 | 0.76 | 0.95 |
| TkeepFirstMARS | 0.02 | 0.15 | 0.03 | 0.02 | 0.02 | 0.23 |
| TkeepSecondMARS | 0.83 | 1.00 | 1.00 | 0.16 | 0.12 | 0.95 |
| WalkingAceleration5 | 1.00 | 1.00 | 0.89 | 0.48 | 1.00 | 0.96 |
| apneaecg | 0.40 | 0.20 | 0.44 | 0.25 | 0.31 | 0.26 |
| apneaecg2 | 0.65 | 1.00 | 0.92 | 1.00 | 1.00 | 0.90 |
| gait1 | 0.18 | 0.36 | 0.53 | 0.51 | 0.48 | 0.47 |
| gaitHunt1 | 0.08 | 0.43 | 0.99 | 0.02 | 0.10 | 0.30 |
| insectEPG2 | 0.12 | 0.23 | 0.73 | 0.14 | 0.81 | 0.96 |
| insectEPG4 | 0.98 | 1.00 | 0.70 | 0.46 | 0.21 | 0.85 |
| lstdbs30791AS | 1.00 | 1.00 | 1.00 | 1.00 | 1.00 | 1.00 |
| mit14046longtermecg | 0.45 | 0.59 | 0.98 | 0.43 | 0.97 | 0.97 |
| park3m | 0.15 | 0.64 | 0.61 | 0.20 | 0.63 | 0.93 |
| qtdbSel1005V | 0.41 | 0.65 | 0.75 | 0.44 | 0.39 | 0.90 |
| qtdbSel100MLII | 0.42 | 0.84 | 0.90 | 0.41 | 0.60 | 0.87 |
| resperation1 | 0.16 | 0.15 | 0.83 | 0.03 | 0.59 | 0.96 |
| s20101mML2 | 0.69 | 0.71 | 1.00 | 0.15 | 0.05 | 0.08 |
| sddb49 | 0.89 | 1.00 | 1.00 | 0.88 | 0.94 | 1.00 |
| sel840mECG1 | 0.41 | 0.66 | 1.00 | 0.32 | 0.28 | 0.36 |
| sel840mECG2 | 0.15 | 0.39 | 0.60 | 0.32 | 0.28 | 0.21 |
| tilt12744mtable | 0.07 | 0.24 | 0.14 | 0.04 | 0.05 | 0.16 |
| tilt12754table | 0.23 | 0.64 | 0.04 | 0.04 | 0.06 | 0.14 |
| tiltAPB2 | 0.92 | 0.98 | 1.00 | 0.36 | 0.83 | 0.38 |
| tiltAPB3 | 0.17 | 0.85 | 0.62 | 0.03 | 0.05 | 0.29 |
| weallwalk | 0.00 | 0.58 | 1.00 | 0.07 | 0.13 | 0.17 |
| **Wins** | 5 | 18 | **24** | 4 | 5 | 12 |
| **Average** | 0.485 | 0.684 | **0.754** | 0.415 | 0.479 | 0.627 |

**Table 16:** Anomaly detection performance across 46 UCR datasets

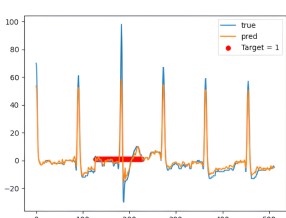 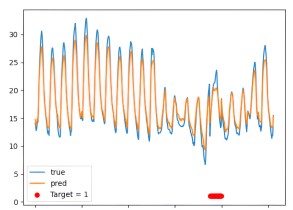 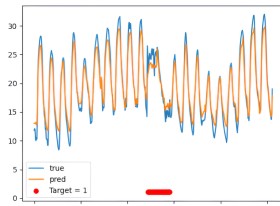

**Figure 15:** UCR Anomaly test set reconstructions visualized, with anomalous regions highlighted for `1sddb40`, `CIMIS44AirTemperature3` and `CIMIS44AirTemperature5`. `true` refers to the ground truth values, while `pred` refers to our reconstruction head's predictions.

### J.2.2 SKOLTECH ANOMALY DETECTION BENCHMARK (SKAB)

**Dataset**   To evaluate our performance on a real world industrial setup, we use the open-source Skoltech Anomaly Benchmark suite Katser & Kozitsin (2020), which consists of a point-wise anomaly detection task using data from 8 sensors attached to a mechanical testbed. The dataset itself consists of 34 sub-dataset instances consisting of both outlier detection, and changepoint detection anomalies.

**Task Description**   We follow the standard setup accompanying the SKAB benchmark for our model, as well as reproducing other baselines on this dataset. This involves splitting the data into several train and test sets, where each instance in the train and test set is trained with a fresh model to reconstruct the training dataset, i.e. minimize MSE on the reconstruction task $||x_{\text{train}} - \hat{x}_{\text{train}}||_2$, and then evaluated by computing the reconstruction of the corresponding test instance, $\hat{x}_{\text{test}}$. Based on the train set reconstruction, we compute the Upper Control Limit (UCL), based on the 99th percentile quantiles, and apply an adjustment factor of $\frac{4}{3}$. Then, for the reconstructed test data $\hat{x}_{\text{test}}$, we classify anomalies if the absolute values of the residuals, i.e. $||x_{\text{test}} - \hat{x}_{\text{test}}||$ lie outside the UCL limit. This exact anomaly detection setup is commonly applied to all baseline models in the test suite.

**Downstream Setup**   Similar to J.2.1, we rely on training a linear head to reconstruct "clean" training data. I.e., we use a single linear layer $\mathbb{R}^{H \times 1}$ to map our embeddings $Z$ back to the raw time series values. The hyperparameters used for training the linear head are listed in 17.

| Hyperparameter | Value |
|---|---|
| Optimizer | AdamW |
| Weight Decay | None |
| Learning Rate | 1e-3 |
| Epochs | 1000 |

**Table 17:** Hyperparameters to train reconstruction head for anomaly detection

**Baselines**   To compare ourselves to a diverse set of models, we include all baselines available in SKAB leaderboard, as well as **T-Rep, TS2Vec, MOMENT**.

The baselines in the SKAB leaderboard come from a diverse set of modeling approaches, which rely on both statistical techniques, as well as more modern CNN/LSTM based methods. We list brief descriptions of the SKAB baselines here:

- **Hotelling's T-squared statistic**: Measures the Mahalanobis distance of new samples from the mean using variances for multivariate process monitoring.
- **Hotelling's T-squared + Q statistic (PCA-based)**: Uses principal component analysis, where $T^2$ captures variation in the principal subspace and $Q$ measures residuals, combined via logical OR for monitoring.
- **Isolation Forest (iForest)**: An ensemble-based method that isolates anomalies as points with short average path lengths in random trees.

- **LSTM-based Neural Network**: An LSTM network trained for anomaly detection using reconstruction error as the anomaly score.

- **Feed-Forward Autoencoder**: A standard autoencoder that detects anomalies via reconstruction error in vector data.

- **Convolutional Autoencoder (Conv-AE)**: A CNN-based autoencoder for anomaly detection in time series via reconstruction error.

- **LSTM Autoencoder (LSTM-AE)**: A sequence-to-sequence LSTM autoencoder that reconstructs temporal patterns and flags anomalies via reconstruction error.

- **LSTM Variational Autoencoder (LSTM-VAE)**: A probabilistic LSTM autoencoder that models latent distributions and detects anomalies using reconstruction error.

- **Variational Autoencoder (VAE)**: A generative model that learns latent variable distributions of input data, with anomalies identified via reconstruction error.

- **MSCRED**: A multi-scale convolutional recurrent encoder-decoder that reconstructs signature matrices of system statuses and uses residuals to detect anomalies.

- **MSET**: A nonparametric statistical modeling technique that estimates values via weighted averages of historical data for anomaly detection.

**Reproducing Baselines**   To ensure a fair comparison, we reproduce the baseline methods `T-Rep`, `TS2Vec`, and `MOMENT` following the protocols described below:

1. **`T-Rep`:** We train the model using the self-supervised contrastive loss used in the paper, on each dataset instance using the official implementation and author-recommended hyperparameters. Subsequently, we append a linear reconstruction head, which is trained using the hyperparameters specified in Table 17. The base encoder remains frozen during this stage.

2. **`TS2Vec`:** We adopt an identical procedure to that of `T-Rep`, i.e., training with the official implementation and hyperparameters, followed by the addition of a frozen base encoder with a trainable linear reconstruction head.

3. **$MOMENT_0$:** We directly evaluate the `AutonLab/MOMENT-1-large` checkpoint in *reconstruction mode*. This configuration utilizes the reconstruction head employed during pretraining and is applied to the test set without any further training or fine-tuning.

4. **$MOMENT_{LP}$:** We employ the same checkpoint in *embedding mode*, in which representations are extracted and paired with a linear reconstruction head. The linear head is trained on the training instances using the hyperparameters from Table 17, consistent with the setup applied to `CHARM`, `T-Rep`, and `TS2Vec`.

Note that the `MOMENT` model was not pre-trained on the SKAB dataset, whereas our model was. This suggests that there may be additional untapped performance potential for `MOMENT` on this benchmark, since pre-training it on SKAB could plausibly improve its results. However, due to the large computational demands of `MOMENT` and our limited access to compute resources, we were unable to conduct this experiment.

**Metrics**   We reported the average F1 score over all instances, as well as the `False Alarm Rate` (FAR) and `Missed Alarm Rate` (MAR), for all baseline models. We outline the mathematical representation of these terms, and their relation to commonly used binary classification metrics here:

$$\text{Missed Alarm Rate (FNR)} = \frac{FN}{TP + FN} = 1 - \text{Recall}$$

$$\text{Specificity (TNR)} = \frac{TN}{TN + FP}$$

$$\text{False Alarm Rate (FPR)} = \frac{FP}{TN + FP} = 1 - \text{Specificity}$$

**Results**   We compile our results on the SKAB benchmark along with the different baseline models collected in separate classes (Self-Supervised vs Classical) in Table 18. Similar to UCR, we here also observe strong performance for CHARM.

**Table 18:** Comparison of anomaly detection performance across baselines and our method. Higher F1 scores are better ($\uparrow$), while lower False Alarm Rate (FAR) and Missed Alarm Rate (MAR) are better ($\downarrow$).

| Category | Method | F1 $\uparrow$ | FAR (%) $\downarrow$ | MAR (%) $\downarrow$ |
|---|---|---|---|---|
| Representation Learning | T-Rep | 0.78 | 12.60 | 28.51 |
| | $MOMENT_0$ | 0.79 | 14.20 | 26.98 |
| | TS2Vec | 0.79 | 12.77 | 27.61 |
| | $MOMENT_{LP}$ | 0.82 | 15.52 | 20.73 |
| Classical | Conv-AE | 0.78 | 13.55 | 28.02 |
| | MSET | 0.78 | 39.73 | 14.13 |
| | T-squared+Q (PCA) | 0.76 | 26.62 | 24.92 |
| | Isolation Forest | 0.29 | **2.56** | 82.89 |
| | LSTM-VAE | 0.56 | 9.13 | 55.03 |
| | MSCRED | 0.36 | 49.94 | 69.88 |
| | **CHARM** | **0.86** | 19.35 | **12.69** |

## J.3 FORECASTING

**Datasets** We evaluated our model on the benchmarks introduced in Autoformer Wu et al. (2021), which have become standard multivariate time series forecasting benchmarks. Specifically, the benchmark suite includes the Electricity Transformer Dataset (ETT), Weather, Exchange Rate and Illness datasets. The model pretraining included the train splits of these datasets (Section H).

The train/valid/test split is identical to the standard protocol in the other baselines we compare with, which is a 6/2/2 split for the ETT datasets, and a 7/1/2 split for all other datasets.

To ensure a fair comparison, we adopt the standard set of lookback horizons and future horizon values across all forecasting datasets, as specified in Table 20. While earlier works primarily use a lookback horizon of 96, more recent studies have also incorporated a longer lookback horizon of 512. To maintain consistency and comparability, we therefore report our linear probing results under both lookback settings. Furthermore, since different papers also employ different prediction horizons, we follow each work's choice of horizons to respect their experimental setup and allow for direct comparison.

**Task Description** Forecasting tasks consist of taking a window of time series data and predicting future time steps. Formally, given an input of dimensions $(T_h \times C)$, where $T_h$ denotes the "lookback" horizon, the goal is to predict the future $T_f$ time steps for all channels.

**Downstream Setup** We use the embeddings of the input horizon data, stack the embeddings across all time steps for each channel, and train the model to minimize an aggregate loss metric[7] between the predicted and true values for each channel.

Since the pretraining task was not designed for direct linear forecasting, nor to produce single-step predictions, we evaluate forecasting performance using the following modeling approaches:

1. **CHARM+LP** A per-dataset, per-channel, per-horizon linear regression head is trained on top of frozen embeddings.

2. **CHARM+NLH:** A common non-linear prediction head is trained across all datasets, channels, and horizons, with the encoder kept frozen.

3. **CHARM+NLH FT:** The full model (encoder + non-linear prediction head) is trained end-to-end, shared across datasets, channels, and horizons.

**Non-Linear Head (NLH)** The head is designed to first mix information across both time and channels, then refine within each channel, and finally project to the forecasting horizon:

- Transformer across time & channels ($n_{\text{heads}} = 4$, $n_{\text{layers}} = 2$, hidden dimension 2048).

---

[7]$\text{loss} = \frac{\text{MSE}+\text{MAE}}{2}$

- Transformer per channel ($n_{\text{heads}} = 4$, $n_{\text{layers}} = 1$, hidden dimension 2048).
- Per-channel linear projection to a maximum horizon of 720.

**It is important to note that in the non-linear head setup, the forecasting module is shared across all datasets, channels, and horizons. The transformer and projection layers are not customized or tuned for any specific dataset, horizon, or channel, ensuring a single common forecasting head is used throughout**. If the target horizon $T_f$ is less than the max horizon (720), we simply apply the loss to the first $T_f$ predictions from the head.

**Training protocol.** All non-linear head models were trained on the full CHARM dataset collection (Section H) without hyperparameter optimization due to resource constraints. The training setup is summarized in Table 19. The linear heads were trained for each (dataset, horizon, channel) combination separately, which is standard for a linear probing setup in time series forecasting, and in line with other baseline implementations. To this end, we conducted hyperparameter optimization as reported in table 21 and present the best results.

| Hyperparameter | Value |
|---|---|
| Lookback horizon | 512 |
| Datasets | All CHARM datasets (Section H) |
| Batch size | 256 (gradients accumulated across datasets) |
| Epoch definition | 10 steps across 4 nodes |
| Max epochs | 60 (early stopping, patience = 5) |
| Optimizer | AdamW |
| Loss | $(\text{MSE} + \text{MAE})/2$ |
| Schedule | Cosine |
| Learning rate | $1 \times 10^{-3}$ |
| Weight decay | 0.01 |

**Table 19: Training protocol for non-linear forecasting (NLH) heads.**

| Dataset | Lookback Horizon $T_h$ | Target Horizon $T_f$ |
|---|---|---|
| ETTh1 | 96/512 | $\{24, 48, 168, 336, 720\}$ |
| ETTh2 | 96/512 | $\{24, 48, 168, 336, 720\}$ |
| ETTm1 | 96/512 | $\{24, 48, 96, 288, 672\}^{\dagger}$ |
| ETTm2 | 96/512 | $\{24, 48, 96, 288, 672\}^{\dagger}$ |
| Weather | 96/512 | $\{96, 192, 336, 672\}$ |
| Exchange Rate | 96/512 | $\{96, 192, 336, 672\}$ |
| Illness | 96/512 | $\{24, 36, 48, 60\}$ |

**Table 20:** Forecasting task specifications. $^{\dagger}$Some papers adopt the same prediction horizons as ETTh1/2 for ETTm1/2.

| Hyperparameter | Value |
|---|---|
| Optimizer | AdamW |
| Weight Decay | [1e-2, 1e-4] |
| Learning Rate | [1e-2, 1e-4] |
| Epochs | 1000 |
| LR Schedule | ReduceLROnPlateau |
| Reduction Factor | 0.1 |
| Early Stopping : Patience | 50 |
| Early Stopping : Tolerance | 1e-6 |

**Table 21: Hyper-parameters to train linear prediction heads for forecasting tasks**

**Baselines** To ensure a fair assessment, we distinguish between three categories of methods: representation learning methods, reconstruction-based methods, and hybrid approaches.

- Representation learning methods focus on extracting meaningful embeddings of the data, independent of the reconstruction objective.

- Reconstruction-based methods emphasize the model's ability to directly predict or reconstruct future values.

- Hybrid approaches combine both ideas: they primarily rely on reconstruction-based training but additionally evaluate the representational power of the learned embeddings.

An important distinction is in the evaluation protocol. Both representation learning methods and hybrid approaches employ linear probing to assess the forecasting power of the embeddings. In contrast, reconstruction-based methods directly evaluate the pretrained model, since their pretraining task is already aligned with forecasting.

To establish a strong baseline, we compare against SOTA foundational time series models from each category. For pure representation learning methods, we include T-REP Fraikin et al. (2024) and TS2Vec Yue et al. (2022). For hybrid methods, we consider `MOMENT` Goswami et al. (2024) and `PatchTST` Nie et al. (2023b). Finally, for reconstruction-based methods, we evaluate `Toto` Cohen et al. (2025), `TIMEMIXER++` Wang et al. (2025), `Moirai` Woo et al. (2024a), and `VisionTS` Chen et al. (2025). We exclude results from works such as `TimesFM` Das et al. (2024), `UniTS` Gao et al. (2024b) and `Chronos` Ansari et al. (2024), as their experimental setups differ substantially from ours, making direct comparison infeasible.

**Reproducing Baselines** We reproduce the baseline methods `T-Rep` and `TS2Vec` on the `Weather`, `ILI`, and `Exchange Rate` datasets following the original papers' pretraining setup. Specifically, each model is first pretrained on the respective dataset, after which a linear forecasting head is added and trained while keeping the base model frozen. The forecasting head is trained using the same architecture and hyperparameters as specified in the original paper's downstream forecasting setup. For the `ETT` datasets, results for both models are taken directly from the original `T-Rep` paper (Fraikin et al., 2024). For reconstruction-based and hybrid models, we report the scores as presented in their respective papers for the corresponding datasets and horizons. The compiled results are shown in Table 23.

**Metrics** We quantitatively assess the model's performance using mean squared error (MSE) and mean absolute error (MAE) metrics averaged over all forecasted time steps and across all target variables, which is standard practice for multivariate forecasting benchmarks.

**Results** The results comparing CHARM with other state-of-the-art representation learning methods, along with the reproduced baselines, are summarized in Table 22. These results underscore the strong performance of CHARM embeddings relative to competing methods in this category. Further, Table 23 demonstrates that CHARM remains competitive with hybrid and reconstruction-based models—including substantially larger models trained on significantly larger datasets (e.g., TOTO, Moirai).

| Dataset | H | T-Rep MSE | T-Rep MAE | TS2Vec MSE | TS2Vec MAE | CHARM+LP MSE | CHARM+LP MAE |
|---|---|---|---|---|---|---|---|
| ETTh1 | 24 | 0.511 | 0.496 | 0.575 | 0.529 | **0.310** | **0.350** |
| | 48 | 0.546 | 0.524 | 0.608 | 0.553 | **0.358** | **0.376** |
| | 168 | 0.759 | 0.649 | 0.782 | 0.659 | **0.451** | **0.430** |
| | 336 | 0.936 | 0.742 | 0.956 | 0.753 | **0.517** | **0.466** |
| | 720 | 1.061 | 0.813 | 1.092 | 0.831 | **0.546** | **0.498** |
| ETTh2 | 24 | 0.560 | 0.565 | 0.448 | 0.506 | **0.186** | **0.267** |
| | 48 | 0.847 | 0.711 | 0.685 | 0.642 | **0.242** | **0.303** |
| | 168 | 2.327 | 1.206 | 2.227 | 1.164 | **0.391** | **0.396** |
| | 336 | 2.665 | 1.324 | 2.803 | 1.360 | **0.430** | **0.427** |
| | 720 | 2.690 | 1.365 | 2.849 | 1.436 | **0.470** | **0.466** |
| ETTm1 | 24 | 0.417 | 0.420 | 0.438 | 0.435 | **0.218** | **0.283** |
| | 48 | 0.526 | 0.484 | 0.582 | 0.553 | **0.282** | **0.324** |
| | 96 | 0.573 | 0.516 | 0.602 | 0.537 | **0.316** | **0.347** |
| | 288 | 0.648 | 0.577 | 0.709 | 0.610 | **0.395** | **0.391** |
| | 672 | 0.758 | 0.649 | 0.826 | 0.687 | **0.482** | **0.441** |
| ETTm2 | 24 | 0.172 | 0.293 | 0.189 | 0.310 | **0.099** | **0.192** |
| | 48 | 0.263 | 0.377 | 0.256 | 0.369 | **0.131** | **0.223** |
| | 96 | 0.397 | 0.470 | 0.402 | 0.471 | **0.172** | **0.253** |
| | 288 | 0.897 | 0.733 | 0.879 | 0.724 | **0.284** | **0.326** |
| | 672 | 2.185 | 1.144 | 2.193 | 1.159 | **0.403** | **0.400** |
| Weather | 96 | 0.195 | 0.280 | 1.672 | 0.904 | **0.158** | **0.199** |
| | 192 | 0.235 | 0.316 | 1.569 | 0.894 | **0.207** | **0.246** |
| | 336 | 0.288 | 0.359 | 2.075 | 1.064 | **0.265** | **0.287** |
| | 672 | 0.362 | 0.402 | 2.828 | 1.305 | **0.347** | **0.340** |
| Exchange Rate | 96 | 1.180 | 0.806 | 0.462 | 0.544 | **0.084** | **0.203** |
| | 192 | 3.947 | 1.344 | 0.968 | 0.765 | **0.182** | **0.302** |
| | 336 | 6.683 | 1.699 | 1.759 | 1.037 | **0.353** | **0.429** |
| | 720 | 3.900 | 1.504 | 2.266 | 1.184 | **0.929** | **0.727** |
| ILI | 24 | 3.631 | 1.227 | 3.463 | 1.173 | **2.799** | **1.080** |
| | 36 | 3.979 | 1.313 | 3.889 | 1.282 | **1.754** | **0.797** |
| | 48 | 4.290 | 1.363 | 4.219 | 1.339 | **1.699** | **0.820** |
| | 60 | 4.361 | 1.375 | 4.198 | 1.329 | **1.740** | **0.838** |

**Table 22:** Representation learning only Long-horizon forecasting results across datasets. Input length = 96. Lower is better. **Bold** = best, Underline = second best. We use a frozen encoder with a linear head for this experiment.

| Dataset | H | Toto MSE | Toto MAE | Moirai_S MSE | Moirai_S MAE | Moirai_B MSE | Moirai_B MAE | Moirai_L MSE | Moirai_L MAE | TimeMixer++ MSE | TimeMixer++ MAE | VisionTS MSE | VisionTS MAE | MOMENT-LP MSE | MOMENT-LP MAE | PatchTST-LP MSE | PatchTST-LP MAE | CHARM+LP MSE | CHARM+LP MAE | CHARM+NLH MSE | CHARM+NLH MAE | CHARM+NLH FT MSE | CHARM+NLH FT MAE |
|---|---|---|---|---|---|---|---|---|---|---|---|---|---|---|---|---|---|---|---|---|---|---|---|
| ETTh1 | 96 | 0.382 | 0.381 | 0.375 | 0.402 | 0.384 | 0.402 | 0.380 | 0.398 | 0.361 | 0.403 | 0.353 | 0.383 | 0.387 | 0.410 | 0.371 | 0.400 | 0.452 | 0.464 | 0.467 | 0.478 | 0.465 | 0.464 |
| | 192 | 0.428 | 0.408 | 0.399 | 0.419 | 0.425 | 0.429 | 0.440 | 0.434 | 0.375 | 0.400 | 0.392 | 0.410 | 0.410 | 0.426 | 0.411 | 0.428 | 0.502 | 0.503 | 0.536 | 0.523 | 0.512 | 0.498 |
| | 336 | 0.457 | 0.422 | 0.422 | 0.429 | 0.450 | 0.456 | 0.514 | 0.474 | 0.416 | 0.423 | 0.407 | 0.423 | 0.422 | 0.437 | 0.445 | 0.446 | 0.565 | 0.540 | 0.600 | 0.564 | 0.560 | 0.530 |
| | 720 | 0.472 | 0.440 | 0.413 | 0.444 | 0.470 | 0.473 | 0.705 | 0.568 | 0.430 | 0.434 | 0.416 | 0.405 | 0.454 | 0.416 | 0.487 | 0.478 | 0.699 | 0.622 | 0.764 | 0.657 | 0.693 | 0.612 |
| ETTh2 | 96 | 0.273 | 0.310 | 0.281 | 0.334 | 0.277 | 0.327 | 0.287 | 0.325 | 0.276 | 0.328 | 0.271 | 0.328 | 0.288 | 0.345 | 0.243 | 0.334 | 0.236 | 0.328 | 0.240 | 0.330 | | |
| | 192 | 0.339 | 0.356 | 0.340 | 0.373 | 0.340 | 0.374 | 0.347 | 0.367 | 0.342 | 0.379 | 0.328 | 0.367 | 0.349 | 0.386 | 0.356 | 0.387 | 0.294 | 0.368 | 0.296 | 0.369 | 0.298 | 0.369 |
| | 336 | 0.410 | 0.387 | 0.362 | 0.393 | 0.371 | 0.401 | 0.377 | 0.393 | 0.346 | 0.398 | 0.345 | 0.381 | 0.425 | 0.377 | 0.395 | 0.434 | 0.334 | 0.394 | 0.332 | 0.391 | | |
| | 720 | 0.375 | 0.400 | 0.380 | 0.416 | 0.394 | 0.426 | 0.404 | 0.421 | 0.392 | 0.415 | 0.388 | 0.422 | 0.403 | 0.439 | 0.395 | 0.434 | 0.424 | 0.448 | 0.421 | 0.446 | 0.395 | 0.432 |
| ETTm1 | 96 | 0.320 | 0.333 | 0.404 | 0.383 | 0.335 | 0.360 | 0.353 | 0.363 | 0.310 | 0.334 | 0.341 | 0.347 | 0.293 | 0.349 | 0.292 | 0.348 | 0.337 | 0.386 | 0.341 | 0.387 | 0.337 | 0.382 |
| | 192 | 0.371 | 0.364 | 0.435 | 0.402 | 0.379 | 0.402 | 0.376 | 0.380 | 0.348 | 0.362 | 0.360 | 0.360 | 0.326 | 0.368 | 0.329 | 0.369 | 0.392 | 0.419 | 0.398 | 0.423 | 0.390 | 0.412 |
| | 336 | 0.408 | 0.388 | 0.462 | 0.416 | 0.394 | 0.416 | 0.399 | 0.395 | 0.376 | 0.391 | 0.377 | 0.374 | 0.352 | 0.384 | 0.364 | 0.391 | 0.434 | 0.442 | 0.437 | 0.449 | 0.434 | 0.440 |
| | 720 | 0.485 | 0.426 | 0.490 | 0.437 | 0.419 | 0.437 | 0.432 | 0.417 | 0.440 | 0.423 | 0.416 | 0.405 | 0.405 | 0.416 | 0.415 | 0.419 | 0.491 | 0.482 | 0.489 | 0.488 | 0.484 | 0.478 |
| ETTm2 | 96 | 0.172 | 0.237 | 0.205 | 0.282 | 0.195 | 0.269 | 0.189 | 0.260 | 0.170 | 0.260 | 0.170 | 0.245 | 0.170 | 0.260 | 0.167 | 0.257 | 0.154 | 0.255 | 0.155 | 0.255 | 0.150 | 0.254 |
| | 192 | 0.232 | 0.280 | 0.318 | 0.261 | 0.303 | 0.300 | 0.247 | 0.300 | 0.229 | 0.291 | 0.262 | 0.305 | 0.227 | 0.297 | 0.229 | 0.300 | 0.188 | 0.282 | 0.197 | 0.287 | 0.186 | 0.283 |
| | 336 | 0.290 | 0.320 | 0.410 | 0.319 | 0.333 | 0.334 | 0.334 | 0.334 | 0.303 | 0.343 | 0.293 | 0.328 | 0.275 | 0.328 | 0.223 | 0.309 | 0.236 | 0.317 | 0.220 | 0.309 | | |
| | 720 | 0.372 | 0.375 | 0.410 | 0.415 | 0.377 | 0.372 | 0.372 | 0.386 | 0.373 | 0.399 | 0.343 | 0.370 | 0.363 | 0.387 | 0.363 | 0.386 | 0.271 | 0.346 | 0.294 | 0.356 | 0.278 | 0.350 |
| Weather | 96 | 0.149 | 0.179 | 0.173 | 0.212 | 0.167 | 0.203 | 0.177 | 0.208 | 0.155 | 0.205 | 0.154 | 0.209 | 0.158 | 0.209 | 0.151 | 0.334? | 0.150 | 0.196 | 0.150 | 0.196 | 0.147 | 0.190 |
| | 192 | 0.192 | 0.223 | 0.216 | 0.250 | 0.209 | 0.241 | 0.219 | 0.249 | 0.201 | 0.245 | 0.244 | 0.275 | 0.197 | 0.248 | 0.203 | 0.249 | 0.197 | 0.240 | 0.198 | 0.239 | 0.191 | 0.232 |
| | 336 | 0.245 | 0.265 | 0.260 | 0.282 | 0.256 | 0.276 | 0.292 | 0.277 | 0.237 | 0.265 | 0.280 | 0.299 | 0.250 | 0.285 | 0.251 | 0.285 | 0.250 | 0.279 | 0.249 | 0.279 | 0.240 | 0.272 |
| | 720 | 0.310 | 0.312 | 0.320 | 0.322 | 0.321 | 0.323 | 0.365 | 0.350 | 0.312 | 0.334 | 0.330 | 0.337 | 0.315 | 0.322 | 0.321 | 0.336 | 0.324 | 0.332 | 0.324 | 0.334 | 0.310 | 0.324 |

**Table 23:** Long-horizon forecasting results across datasets. Input length = 512. Lower is better. **Bold** = best, Underline = second best. Last three columns are our CHARM variants; *CHARM+LP* is a **Rep.** approach, while *CHARM + NLH* and *CHARM + NLH FT* are **Rec.** approaches.

**Visualizations** We present a sample of forecasting results from **CHARM+LP** and using a lookback window of 96.

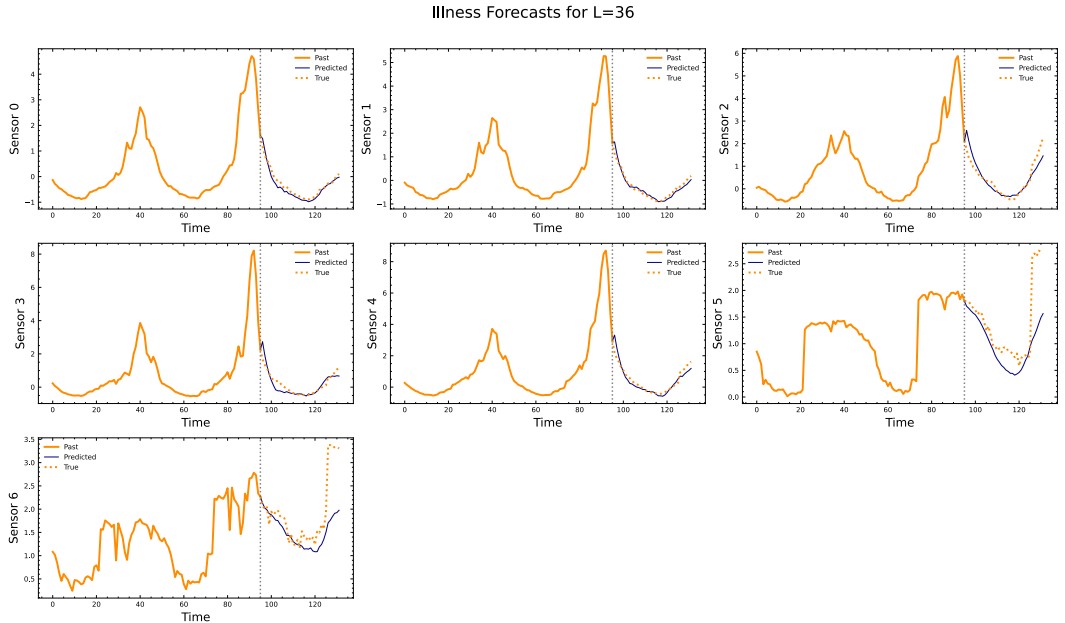

**Figure 16:** Illness Forecasts

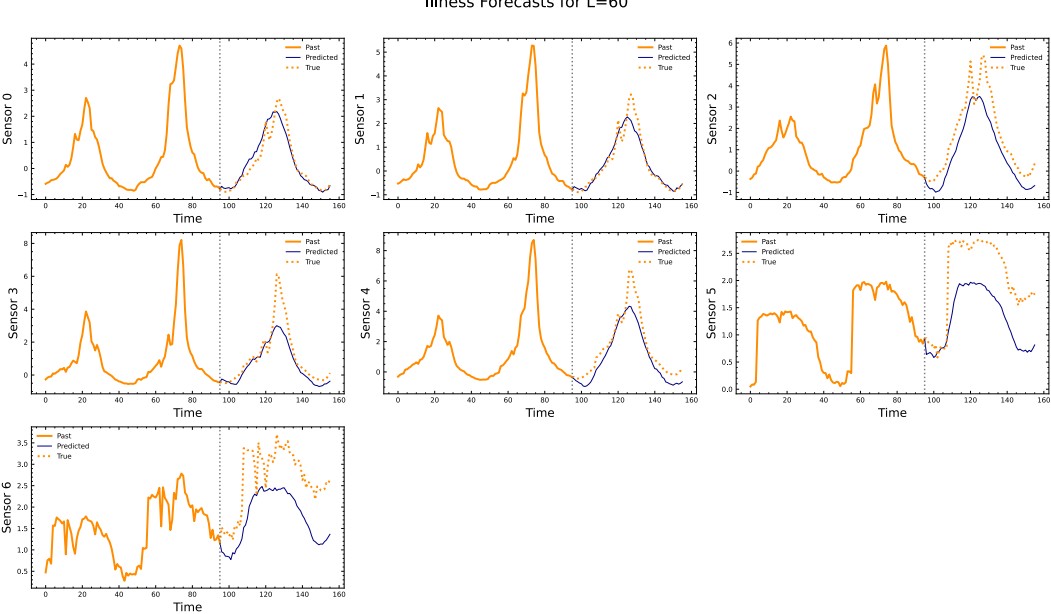

**Figure 17:** Illness Forecasts

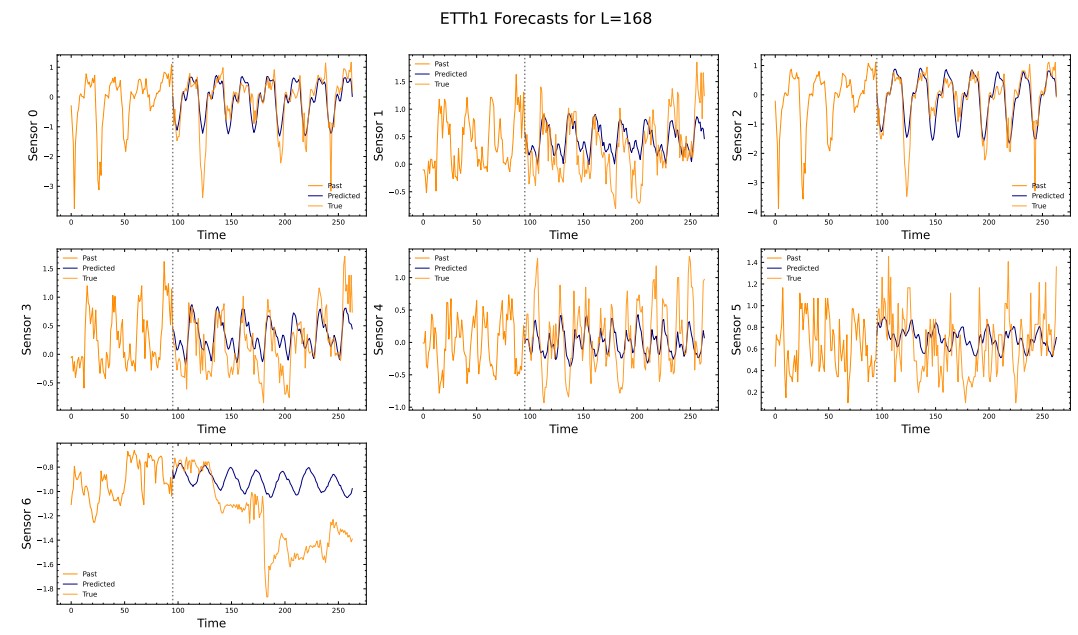

**Figure 18:** ETTh1 Forecasts

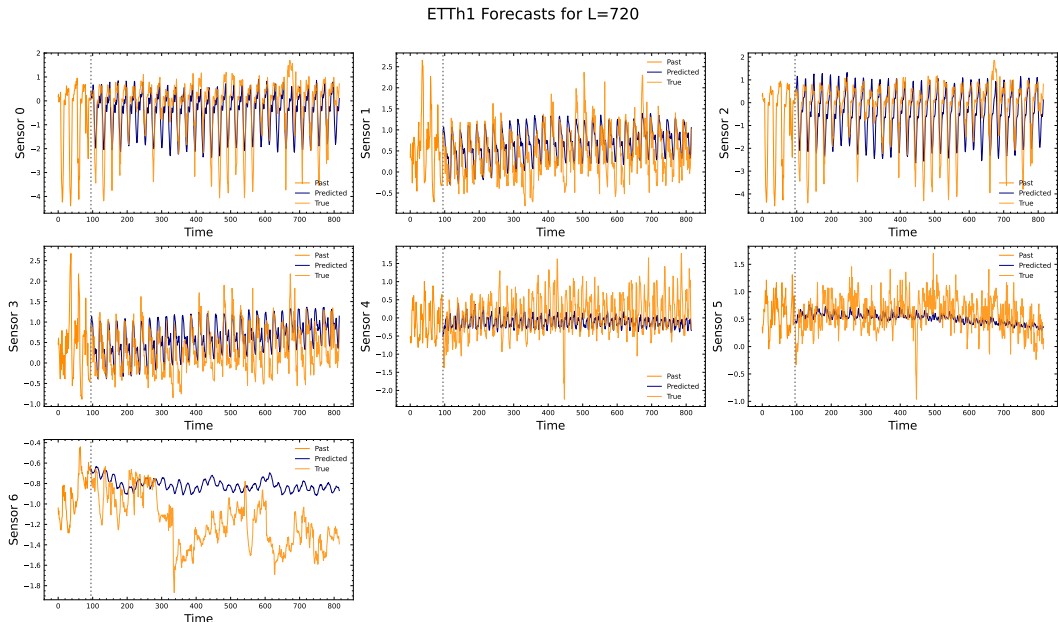

**Figure 19:** ETTh1 Forecasts

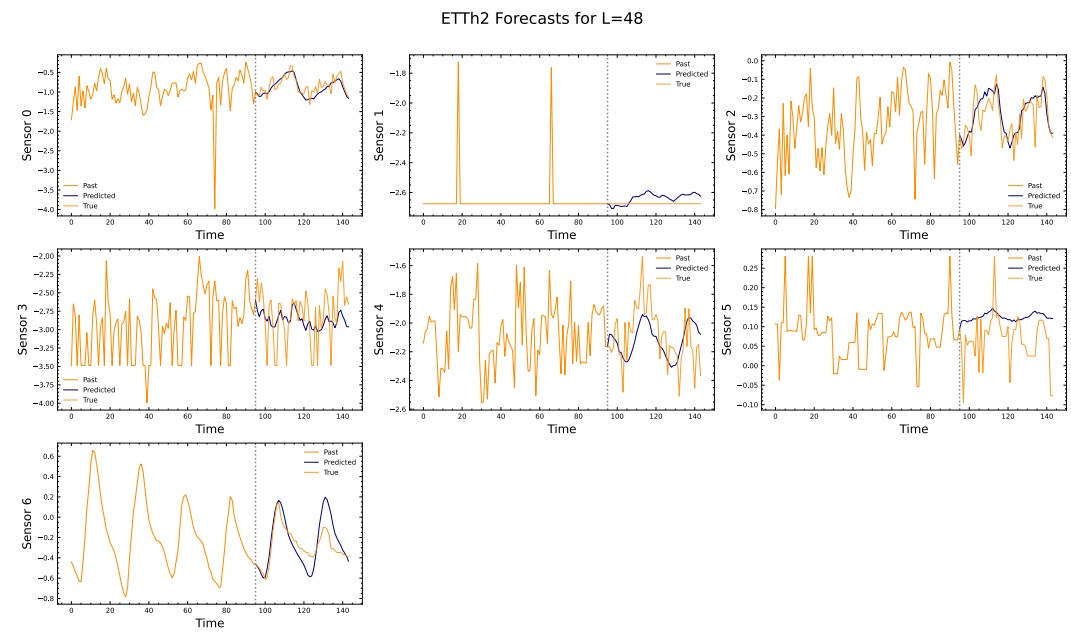

**Figure 20:** ETTh2 Forecasts

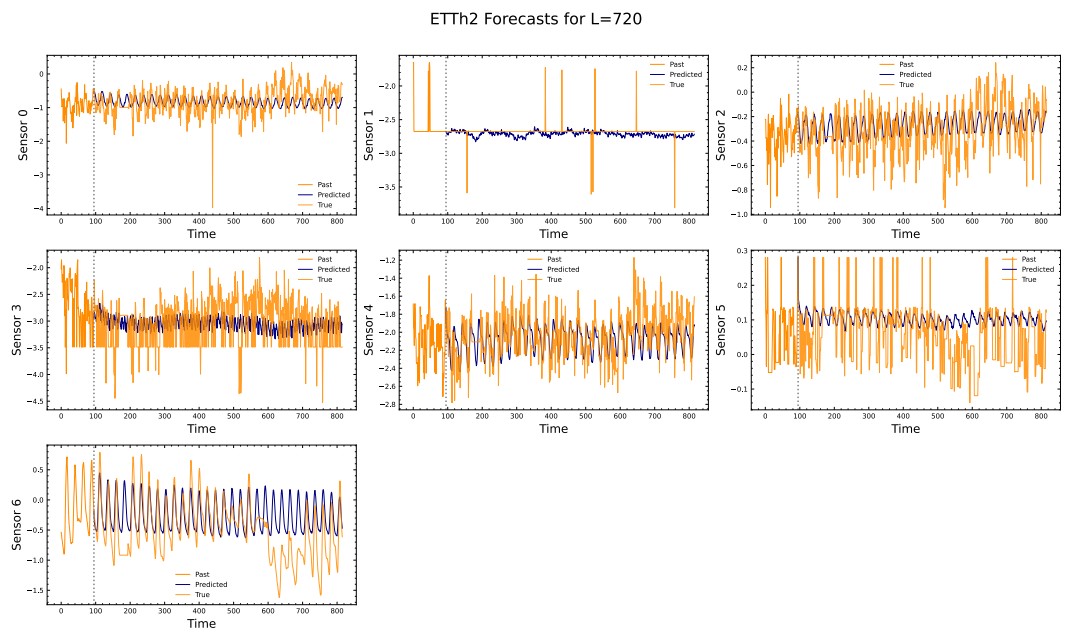

**Figure 21:** ETTh2 Forecasts

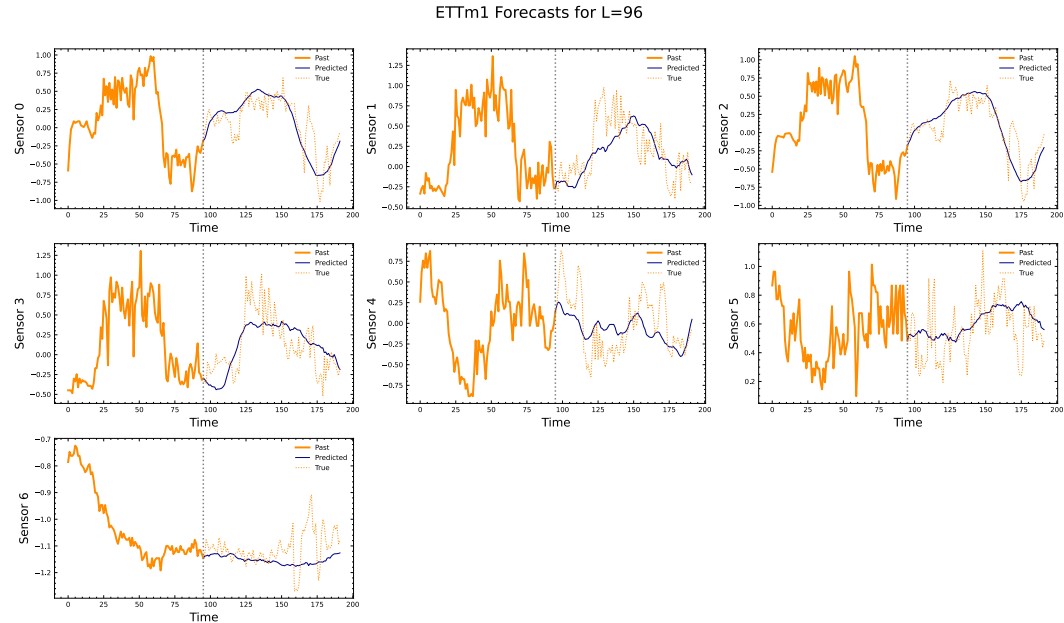

**Figure 22:** ETTm1 Forecasts

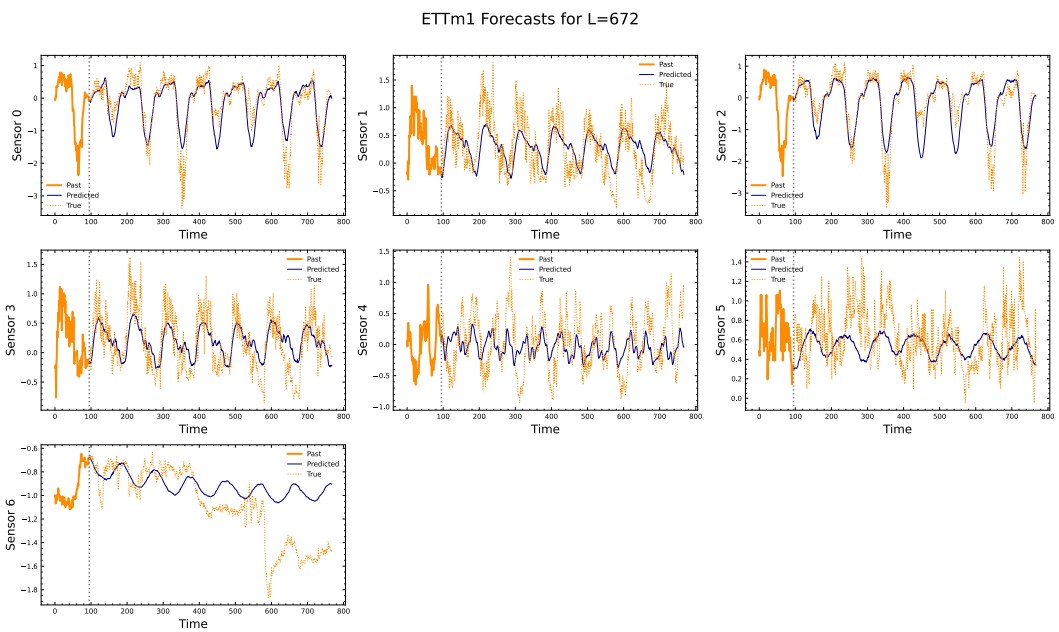

**Figure 23:** ETTm1 Forecasts

**Linear Probing and Pooling Ablations** Experiments related to pooling strategies, shown in Table 24, were designed to contrast different pooling approaches commonly used to aggregate data in other domains and to examine their applicability to time series. We also experimented with an MLP to investigate whether the embeddings benefit more from a non-linear model, since the training objective is not aimed at maximizing linear predictability. We only compare these methods using the ETTh1/2 datasets with a lookback window of 96 and forecasting horizons of $24, 48, 168, 336, 720$.

To further study the effect of different pooling strategies, we evaluated the following three approaches:

1. **Flattening**: The embeddings are flattened into a single vector representation for each channel.

$$Z : \mathbb{R}^{T \times H} \to Z_{\text{flat}} \in \mathbb{R}^{TH}$$

2. **Mean Pooling**: The embeddings are averaged over the time dimension (but not across channels), yielding an $H$-dimensional representation per channel.

$$Z : \mathbb{R}^{T \times H} \to Z_{\text{mean}} \in \mathbb{R}^{H}$$

3. **Last Time Step**: The embedding from the last time step of each channel is taken as the representative embedding.

$$Z : \mathbb{R}^{T \times H} \to Z_{-1} \in \mathbb{R}^{H}$$

In addition, we experimented with:

- **Frozen Encoder + 2-Layer MLP**: A per-dataset, per-channel, per-horizon MLP head (two linear layers with ReLU activations) trained on top of frozen embeddings.

| Dataset | Pool | Head | 24 | | 48 | | 168 | | 336 | | 720 | |
|---|---|---|---|---|---|---|---|---|---|---|---|---|
| | | | MSE | MAE | MSE | MAE | MSE | MAE | MSE | MAE | MSE | MAE |
| ETTh1 | **none**[1a] | linear[1b] | **0.31** | **0.35** | 0.36 | **0.38** | 0.45 | **0.43** | 0.52 | 0.47 | 0.55 | **0.50** |
| | | MLP[2b] | 0.32 | 0.36 | **0.37** | 0.38 | **0.46** | 0.44 | **0.50** | **0.46** | **0.54** | **0.50** |
| | **last time step**[2a] | linear[1b] | 0.39 | 0.39 | 0.43 | 0.41 | 0.51 | 0.46 | 0.53 | 0.47 | 0.54 | 0.50 |
| | | MLP[2b] | **0.37** | 0.38 | 0.41 | 0.40 | 0.49 | 0.45 | 0.54 | 0.48 | 0.55 | 0.51 |
| | **mean**[3a] | linear[1b] | 0.66 | 0.49 | 0.68 | 0.50 | 0.72 | 0.53 | 0.69 | 0.54 | 0.69 | 0.57 |
| | | MLP[2b] | 0.61 | 0.50 | 0.74 | 0.51 | 0.77 | 0.54 | 0.74 | 0.55 | 0.73 | 0.58 |
| ETTh2 | **none**[1a] | linear[1b] | **0.19** | **0.27** | **0.24** | **0.30** | **0.39** | **0.40** | **0.43** | **0.43** | **0.47** | **0.47** |
| | | MLP[2b] | 0.20 | **0.27** | 0.25 | 0.31 | 0.40 | **0.40** | 0.46 | 0.44 | 0.51 | 0.48 |
| | **last time step**[1a] | linear[1b] | **0.19** | 0.28 | 0.25 | 0.31 | 0.40 | **0.40** | 0.44 | **0.43** | 0.49 | 0.48 |
| | | MLP[2b] | 0.20 | 0.28 | 0.26 | 0.32 | 0.40 | **0.40** | 0.45 | 0.44 | 0.49 | 0.48 |
| | **mean**[1a] | linear[1b] | 0.25 | 0.32 | 0.29 | 0.35 | 0.44 | 0.43 | 0.44 | 0.45 | 0.50 | 0.49 |
| | | MLP[2b] | 0.25 | 0.33 | 0.30 | 0.35 | 0.44 | 0.43 | 0.46 | 0.45 | 0.50 | 0.49 |

**Table 24:** Ablation results comparing pooling strategies and heads across ETTh1 and ETTh2. **Bolded** values denote best within each dataset and horizon. The encoder is kept frozen for this experiment.

From Table 24, we observe that no pooling (1a) combined with either a linear or MLP probe yields the best results on both ETTh1 and ETTh2. Interestingly, we observe that for ETTh1 using just the last time step's embedding (1b) yields competitive scores with an average increase of 8.7% (MSE) and 4.2% (MAE) when compared to no pooling (1a). Comparatively, mean pooling (1c) has an increase of 60.5% (MSE) and 24.4% (MAE) Similarly, for ETTh2, we observe that using the last time step embeddings (1b) has only a 0.85% (MSE) and 1.33% (MAE) increase in error, when compared to mean pooling which has a 9.32% (MSE) and 8.49% (MAE) increase in error.

This observation is in line with (Bardes et al., 2024), which demonstrated that using attentive probing to pool embeddings was empirically superior for downstream task performance compared to directly mean pooling the embeddings, which can potentially result in lossy, diffuse representations which fail to capture finer granularities in the data.

## K ABLATIONS

To better understand the effects of different components in our model, we perform a series of ablations involving the proposed architectural additions - (i) TCN featurization layer, (ii) text based attention mechanisms, (iii) effect of description quality, (iv) effect of embedding model, and (v) alternative approaches to multi modal text + time series. To quantify the effect of each of these changes, we measure performance on forecasting and classification by measuring the following quantities:

1. Total number of correct classifications for UEA

2. Average accuracy for UEA

3. Mean Squared Error for ETTh1 ($T_f = 168$)

4. Mean Absolute Error for ETTh1 ($T_f = 168$)

5. Mean Squared Error for ETTh2 ($T_f = 168$)

6. Mean Absolute Error for ETTh2 ($T_f = 168$)

For the ablation study, we train and probe the model with the following protocols:

**Pretraining**  The base model is pretrained with a subset of datasets (UEA, ETTh, ETTm, Weather, Illness), for 50 epochs, with a learning rate of 5e-4.

**Classification Evaluations**  The hyperparameters used for measuring classification performance are listed in Table 25. We use mean-pooled embeddings, i.e. $\bar{Z} \in \mathbb{R}^{B \times H}$, instead of flattened embeddings, $Z \in \mathbb{R}^{B \times T \times C \times H}$.

**Forecasting Evaluations**  The forecasting setup is identical to the frozen linear model setup used in the downstream forecasting task. The hyperparameters are listed in Table 21. The embeddings are flattened for all timesteps and passed to a linear layer.

| Hyperparameter | Value |
|---|---|
| C | {0.0001, 0.1, 1000} |
| kernel | {'rbf'} |
| degree | {3} |
| gamma | {'scale'} |
| coef0 | {0} |
| shrinking | {True} |
| probability | {False} |
| tol | {0.001} |
| cache_size | {200} |
| class_weight | {None} |
| verbose | {False} |
| max_iter | {10000000} |
| decision_function_shape | {'ovr'} |
| random_state | {None} |

Table 25: SVM Hyperparameter Grid for Ablations

The ablation results for (i) and (ii) can be found in the main text in Table 5.

We cover the setup for the remaining ablations (iii), (iv), and (v) here, along with their results.

- **(iii) Effect of description quality**
  As channel descriptions are a first class citizen in training CHARM, we perform an ablation to investigate the effect of description quality on the model's performance. To this end we consider three cases:
  1) **Annotated descriptions**: manually curated sensor descriptions obtained from the official dataset metadata. These are obtained through either manual human annotation obtained by parsing the accompanying dataset metadata files, or are natively provided by the dataset provider.
  2) **Noisy descriptions**: high quality annotated descriptions, but with words dropped at random (with $p = 0.2$) during both training and evaluation.
  3) **Ordinal descriptions**: replace the annotated descriptions with structured, placeholder descriptions: [Sensor1, Sensor2, Sensor3...] for all datasets.

- **(iv) Effect of text embedding model**
  We investigate the usage of different embedding models to assess the effect on downstream performance. We use 1) nomic (Nussbaum et al., 2025), 2) minilm (Wang et al., 2020), and 3) mpnet (Song et al., 2020) as representative models to assess the downstream impact on scores.

- **(v) Alternative multi-modal approaches**

  To investigate how naive multimodal approaches compare to our setup, we remove the text based layers altogether, and simply add in the channel description embeddings (from an LLM embedding model) in a pointwise sense to the time series embeddings. This is analogous to adding in position embeddings in the first layer of a vanilla transformer (Vaswani et al., 2017). We investigate adding these solely in the first layer, as well as in all layers. For numerical stability, we apply a `LayerNorm` on these embeddings to ensure they are of an appropriate scale.

| Configuration | # Correct | Accuracy | $\text{ETTh1}_{168}$ MSE | $\text{ETTh1}_{168}$ MAE | $\text{ETTh2}_{168}$ MSE | $\text{ETTh2}_{168}$ MAE |
|---|---|---|---|---|---|---|
| Ordinal descriptions | 4792 | 70.3% | 0.52 | 0.55 | 0.59 | 0.83 |
| Noisy descriptions | 4813 | 71.2% | 0.44 | 0.48 | 0.59 | 0.85 |
| **w/ annotated descriptions** | **4897** | **71.4%** | **0.42** | 0.49 | **0.57** | **0.80** |

**Table 26: Effect of Sensor Descriptions (TCN$_{\text{conv}}$)**

| Embedding Model | # Correct | Accuracy | $\text{ETTh1}_{168}$ MSE | $\text{ETTh1}_{168}$ MAE | $\text{ETTh2}_{168}$ MSE | $\text{ETTh2}_{168}$ MAE |
|---|---|---|---|---|---|---|
| mpnet | 4893 | 71.3% | **0.41** | **0.45** | 0.63 | 0.85 |
| minilm | **4902** | **72%** | 0.43 | 0.47 | 0.65 | 0.95 |
| nomic | 4897 | 71.4% | 0.42 | 0.49 | **0.57** | **0.80** |

**Table 27: Effect of Text Embedding Models**

| Configuration | # Correct | Accuracy |
|---|---|---|
| w/ additive embeddings (`all layers`) | 4095 | 60.5% |
| w/ additive embeddings (`layer 0`) | 4375 | 63.3% |
| **$\Delta + G$** | **4897** | **71.4%** |

**Table 28: Alternative Multimodal Approaches: additive vs. custom attention**

As shown in Table 26, perturbing or replacing channel descriptions leads to a moderate performance drop; however, the model remains reasonably robust to noisy descriptions. Table 27 reveals varied and inconclusive trends across different textual embedding models, with each performing well on distinct metrics. Finally, Table 28 suggests that the naive integration of text embeddings into the architecture is overly heavy-handed and results in performance degradation, particularly when embeddings are injected directly into all layers of the encoder stack. These findings indicate that incorporating channel descriptions into a time series transformer requires greater nuance and more principled design choices.

## L  EMBEDDING VISUALIZATIONS

### L.1  EMERGENCE OF INTRA-CLASS LABEL SEPARATION

To analyze how our model's embeddings evolve over training, we plot similarity heatmaps of our embeddings on labelled datasets.

We first obtain embeddings for a dataset by sampling a subset (approximately 50 samples) of the full dataset, while ensuring we have full label coverage. Given this embedding matrix $\mathbf{Z} \in \mathbb{R}^{N_t \times T \times C \times H}$, we obtain our mean-pooled embeddings $\bar{\mathbf{Z}} \in \mathbb{R}^{N_t \times H}$ by averaging over the channel and time dimension.

Finally, the $N_t \times N_t$ similarity matrix, $\mathbf{S}$ is obtained as follows:

$$\mathbf{S}_{i,j} = ||\mathbf{Z}_{i,:} - \mathbf{Z}_{j,:}||_1 \tag{11}$$

We visualize the similarity matrix as a heatmap, as shown in Figures 24 to 26, and observe the emergence of structured clusters aligned with class labels. As training progresses, a block-diagonal structure[8] becomes increasingly prominent, wherein samples sharing the same label exhibit reduced Euclidean separation compared to those from different classes. This pattern reflects a progressive tightening of intra-class representations, indicative of improved semantic organization in the learned embedding space.

---

[8] The heatmaps have a block structure because the labels are grouped together on each axis before plotting.

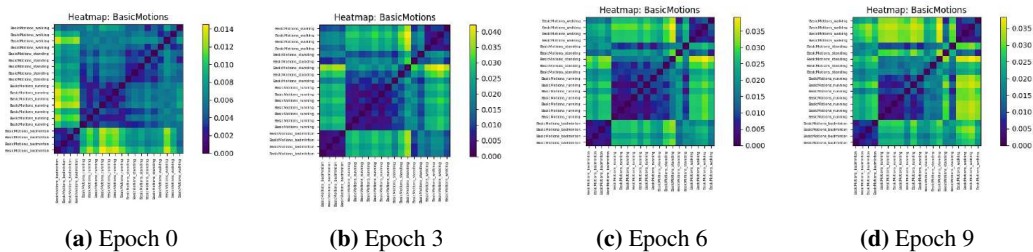

**(a)** Epoch 0      **(b)** Epoch 3      **(c)** Epoch 6      **(d)** Epoch 9

**Figure 24:** Evolution of `BasicMotions` similarity heatmaps over training epochs

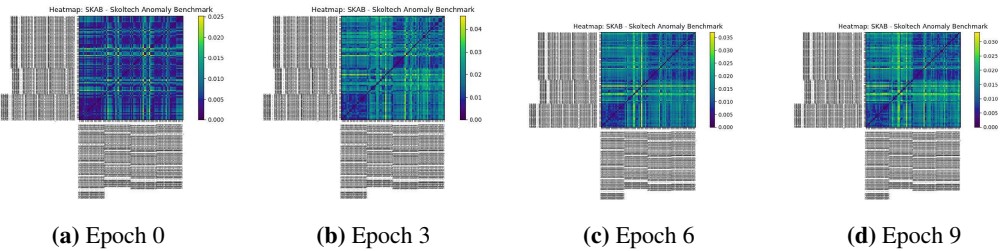

**(a)** Epoch 0      **(b)** Epoch 3      **(c)** Epoch 6      **(d)** Epoch 9

**Figure 25:** Evolution of `Skoltech Anomaly Benchmark` similarity heatmaps over training epochs

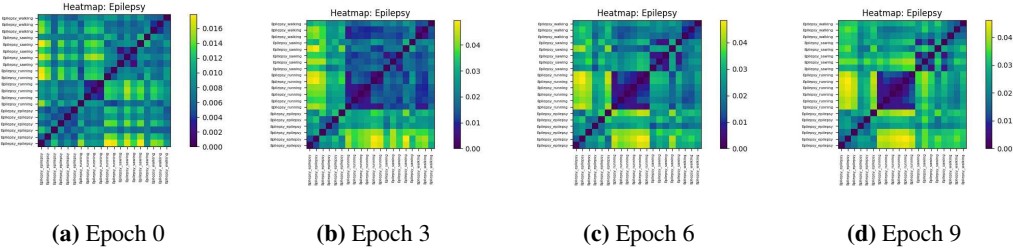

**(a)** Epoch 0      **(b)** Epoch 3      **(c)** Epoch 6      **(d)** Epoch 9

**Figure 26:** Evolution of `Epilepsy` similarity heatmaps over training epochs

## L.2 EVOLUTION OF CHANNEL GATES

In this section we aim to visualize how channel gates, as defined in Paragraph Section 2.1.1, evolve over the course of training our model. We plot the gating matrix, $\mathbf{G}_d$, for each dataset for different checkpoints.

As illustrated in Figure 28, the inter-channel gating mechanism enables the model to dynamically modulate attention across channels, selectively emphasizing or suppressing information based on configurations that minimize the self-supervised learning (SSL) loss. We also empirically observe that the regularization loss begins to increase after an initial decline which suggests that after a certain point the model's embeddings require richer contextual information to continue improving.

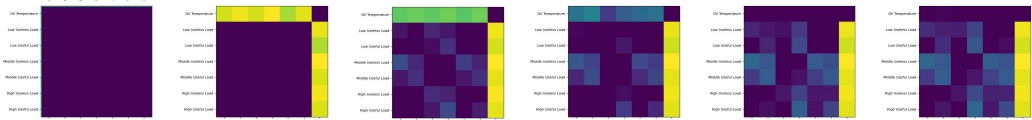

**Figure 27:** Evolution of Channel Gates for the `ETT` Dataset

The `ETT` dataset introduced by (Zhou et al., 2021a) comprises seven variables: `High Useful Load, Middle Useful Load, Low Useful Load, High Useless Load, Middle`

`Useless Load`, `Low Useless Load`, and `Oil Temperature`. Among these, `Oil Temperature` serves as the target variable, with the remaining six acting as input features. During training, we observe a notable evolution in the learned channel gating patterns. Initially, the `Oil Temperature` channel does not attend to any other inputs, as indicated by its high gating values across all dimensions in Figure 27. However, as training progresses, this channel begins to incorporate information from all other variables. Interestingly, this behavior is asymmetric: while the target channel attends to all input features, the reverse does not occur—the other channels do not attend to `Oil Temperature`. This asymmetry manifests as a distinctive row-column pattern in the gating matrix and aligns with the underlying data semantics, where the target variable is causally influenced by the independent variables but not vice versa. These observations suggest that introducing learnable gating mechanisms can reveal interpretable, directional dependencies between variables which also increases model interpretability.

**Figure 28:** Evolution of inter channel gates during training. Checkpoints extracted at `epoch=0;step=49`, `epoch=0;step=499`, `epoch=0;step=999`, `epoch=2;step=49`, `epoch=6;step=49`, `epoch=8;step=49`.
Each row represents a particular dataset. Each column represents a sampled checkpoint as training progresses. Each heatmap represents $\mathbf{G}_d$ for a particular dataset, which is a $C \times C$ matrix with values in $[0, 1]$. Brighter colors on the heatmap represent **higher** gating values, i.e. decreased cross-channel interactions.

