# OpenReview forum: "Unlocking Time Series Foundation Models with Channel Descriptions"
_ICLR.cc/2026/Conference — ICLR 2026 Conference Withdrawn Submission_

### Official Review · Reviewer_pN1R · 2025-10-28

**Soundness:** 2
**Presentation:** 1
**Contribution:** 2
**Rating:** 4
**Confidence:** 3

**Summary:**

This paper proposes a neural network architecture and self-supervised training method for multivariate time series, CHARM, that leverages the textual information of each channel. The paper designs a text-aware TCN and self-attention network, and uses the Joint Embedding Prediction Architecture (JEPA) for model training. The trained model can be used for three downstream tasks: prediction, classification, and anomaly detection. Experiments in the paper demonstrate that integrating channel descriptions significantly improves representation quality.

**Strengths:**

1. The experiments comprehensively demonstrate the performance of the proposed CHARM.
2. Using JEPA as the self-supervised learning objective function is somewhat novel.

**Weaknesses:**

1. The idea of integrating textual information into time series analysis demonstrates a certain level of performance improvement and has been explored in existing research. However, it faces the challenge of insufficient training data, which is inadequate for training a robust foundational model. Many open-source multivariate time series datasets lack descriptive information for corresponding columns. Moreover, current TSFMs (e.g., Chronos and Moirai) heavily rely on synthetic data, which also lacks column descriptions.
2. The writing of the paper requires significant improvement. The paper involves extensive mathematical notation with unclear descriptions. Some concepts are defined but not utilized, such as $\bar{\Delta}$ on line 234. Additionally, the font in Figure 1 is too small to read.
3. The experiments primarily focus on small-scale datasets and lack validation on larger benchmarks, such as Gitf-Eval.

**Questions:**

1. In line 291, the authors mention that $\mathcal{I}_{\textrm{ctx}}$ and another two symbols are used to represent subsets of $\textbf{pos}$. By definition, these sets appear to be used for selecting time series variables. However, why do the authors refer to them as "time indices"? Subsequent descriptions also indicate they are actually used for selecting data points along the temporal axis (i.e., past and future data).
2. How did the authors construct a large-scale dataset with column description information for training this foundational model?

---

### Official Review · Reviewer_rNsA · 2025-10-29

**Soundness:** 1
**Presentation:** 1
**Contribution:** 2
**Rating:** 2
**Confidence:** 5

**Summary:**

This paper introduces a model that improves representation quality for multivariate time series by incorporating channel-level textual descriptions. This design enables the model to exploit contextual information associated with individual sensors while remaining invariant to the order of the channels. Extensive experiments on three classic time series tasks show the performance of the proposed method.

**Strengths:**

- This paper is easy to follow.
- Incorporates textual channel descriptions into time-series modeling, enhancing semantic understanding and generalization.
- The proposed approach achieves competitive performance across forecasting, classification, and anomaly detection tasks, often surpassing existing SOTA baselines.

**Weaknesses:**

- The authors’ claim that Transformer-based architectures for time series representation learning remain limited is debatable. Recent studies have shown strong Transformer variants for time series tasks (e.g., [1]). The paper would benefit from a more balanced discussion acknowledging these developments.

- The code and pre-trained models are not yet available, making it difficult to verify the results and reproduce the experiments.

- Figure 6 lacks a clear explanation. Each subfigure should be labeled and described to improve interpretability. The interpretability of how textual descriptions influence specific channels or predictions remains underexplored.

1. https://github.com/thuml/Time-Series-Library

**Questions:**

Please see the Weakness.

---

### Official Review · Reviewer_H2Rv · 2025-10-30

**Soundness:** 2
**Presentation:** 2
**Contribution:** 2
**Rating:** 4
**Confidence:** 3

**Summary:**

This paper presents CHARM, a new foundational model designed to improve the representation of multivariate time series, with applications in forecasting, classification, and anomaly detection. The key innovation of CHARM lies in its integration of channel descriptions and their interactions within the model architecture. Specifically, the authors extract textual embeddings from the channel descriptions, which are then combined with a temporal convolution network that captures temporal dynamics. Additionally, they introduce a custom cross-attention mechanism to enhance channel correlation. For training, the model uses a self-supervised learning approach, similar to JEPA, tailored for time series analysis.

**Strengths:**

1. Proposing a versatile foundational model for time series that performs well in forecasting, classification, and anomaly detection is both timely and important.

2. The results presented are promising, with the model outperforming existing competitors.

**Weaknesses:**

1. The methodology feels somewhat empirical, especially regarding the importance of channel descriptions, which is a key innovation. It's unclear whether channel descriptions actually improve performance. The authors should provide examples where this is critical and conduct experiments (even synthetic ones) to demonstrate the benefit of channel descriptions. The same applies to the cross-attention mechanism its value in modeling channel correlations needs better motivation and clear benchmarking, especially on datasets where channel dependencies are significant. Additionally, the use of the JEPA training paradigm should be better justified for time series, with clear reasoning or experiments to support its applicability.

2. It would be useful to show standard deviations and conduct statistical tests. A discussion on the computational cost of the approach is also recommended.

**Questions:**

1. On motivation, is the claim about the importance of channel descriptions for forecasting justified? The authors should provide examples where channel descriptions intuitively improve forecasting. Additionally, they should demonstrate how their architecture effectively incorporates this information. A controlled experiment showing how performance (e.g., MSE) improves as more informative channel descriptions are added would strengthen this claim.

2. The same approach should be applied to channel interdependency. In a controlled setting with known dependencies, the authors could test how well the method reduces MSE as dependency strength increases, and track how attention weights change. This would provide clearer evidence of the architecture's value beyond real-world datasets.

3. A discussion on computational complexity and cost would also be valuable.

4. Please ensure the article is self-contained and defines all key concepts. For instance except mistakes from my side, the triplet (T, D, pos) is not clearly defined in the main text, leaving readers to refer to the appendix for clarification. Specifically, while T is understood as the time series, D and pos (description and position) are not defined upfront. Make sure all concepts are clearly explained within the main text.

---

### Official Review · Reviewer_WqLj · 2025-10-30

**Soundness:** 3
**Presentation:** 2
**Contribution:** 3
**Rating:** 6
**Confidence:** 2

**Summary:**

The paper introduces CHARM (Channel-Aware Representation Model), a foundation model for multivariate time series that integrates channel-level textual descriptions into its architecture to enrich semantic representations. By combining a description-aware temporal encoder, channel-offset attention with gating, and a JEPA-based self-supervised objective, CHARM enables robust, noise-resistant, and semantically grounded representation learning across diverse downstream tasks such as forecasting, classification, and anomaly detection. Extensive experiments demonstrate its versatility and generalization ability, and its reproducibility statement strengthens scientific transparency. Overall, the work offers an innovative step toward bridging textual semantics and time-series modeling, though further clarification on implementation details, scalability, and differentiation from recent JEPA-based methods would enhance its contribution.

**Strengths:**

The paper proposes a novel and conceptually strong framework that unites textual and numerical modalities in time-series modeling. By integrating channel-level textual descriptions into a JEPA-based architecture, CHARM enables semantically grounded and noise-resilient representations that reflect real-world interpretability. The use of description-aware temporal encoding and channel-offset attention demonstrates a creative approach to modeling cross-channel dependencies without relying on explicit supervision. The framework’s broad evaluation across forecasting, classification, and anomaly detection tasks showcases its versatility, while the emphasis on reproducibility adds credibility and scientific rigor.

**Weaknesses:**

The paper’s novelty claims are not sufficiently distinguished from recent JEPA-based or multimodal foundation models such as “Time to Embed: Unlocking Foundation Models for Time Series with Channel Descriptions” and “Joint Embeddings Go Temporal”. The implementation details are under-specified, which hinders reproducibility. Furthermore, the discussion of scalability and robustness is somewhat superficial; the model’s dependence on the quality of channel descriptions and its computational cost for long sequences remain unclear.

**Questions:**

1. How does CHARM’s JEPA objective technically differ from prior time-series JEPA applications?
2. What specific mechanisms make the model robust when channel descriptions are noisy or unavailable?
3. How is scalability addressed for very long multivariate sequences?
4. Could you provide sensitivity or ablation studies for the control weights (λ₁, λ₂) and multi-resolution losses?
5. How do the learned embeddings compare qualitatively to contrastive SSL baselines in terms of interpretability and stability?

---

### Note · Authors · 2025-12-10

I have read and agree with the venue's withdrawal policy on behalf of myself and my co-authors.